# Adaptive wavelet distillation from neural networks through interpretations

**Wooseok Ha**[1]
haywse@berkeley.edu

**Chandan Singh**[2]
chandan_singh@berkeley.edu

**Francois Lanusse**[3]
francois.lanusse@cea.fr

**Srigokul Upadhyayula**[4]
sup@berkeley.edu

**Bin Yu**[1,2]
binyu@berkeley.edu

[1] Statistics Department, UC Berkeley
[2] EECS Department, UC Berkeley
[3] AIM, CEA, CNRS; Université Paris-Saclay, Université Paris Diderot, Sorbonne Paris Cité
[4] Advanced Bioimaging Center, Department of Molecular & Cell Biology, UC Berkeley

## Abstract

Recent deep-learning models have achieved impressive prediction performance, but often sacrifice interpretability and computational efficiency. Interpretability is crucial in many disciplines, such as science and medicine, where models must be carefully vetted or where interpretation is the goal itself. Moreover, interpretable models are concise and often yield computational efficiency. Here, we propose adaptive wavelet distillation (AWD), a method which aims to distill information from a trained neural network into a wavelet transform. Specifically, AWD penalizes feature attributions of a neural network in the wavelet domain to learn an effective multi-resolution wavelet transform. The resulting model is highly predictive, concise, computationally efficient, and has properties (such as a multi-scale structure) which make it easy to interpret. In close collaboration with domain experts, we showcase how AWD addresses challenges in two real-world settings: cosmological parameter inference and molecular-partner prediction. In both cases, AWD yields a scientifically interpretable and concise model which gives predictive performance better than state-of-the-art neural networks. Moreover, AWD identifies predictive features that are scientifically meaningful in the context of respective domains. All code and models are released in a full-fledged package available on Github. [1]

## 1 Introduction

Recent advancements in deep learning have led to impressive increases in predictive performance. However, the inability to interpret deep neural networks (DNNs) has led them to be characterized as black boxes. It is often critical that models are inherently interpretable [1–3], particularly in high-stakes applications such as medicine, biology, and policy-making. In these cases, interpretations which are relevant to a particular domain/audience [3] can ensure that models behave reasonably, identify when models will make errors, and make the models more amenable to inspection and improvement by domain experts. Moreover, interpretable models tend to be faster and more computationally efficient than large neural networks.

---

[1] ⭘ github.com/Yu-Group/adaptive-wavelets

35th Conference on Neural Information Processing Systems (NeurIPS 2021).

One promising approach to constructing interpretable models without sacrificing prediction performance is model distillation. Model distillation [4–6] transfers the knowledge in one model (i.e., the teacher), into another model (i.e., the student), where the student model often has desirable properties, such as being more interpretable than the teacher model. Recent works have considered distilling a DNN into inherently interpretable models such as a decision tree [7–9] or a global additive model [10], with some success. Here, we consider distilling a DNN into a learnable wavelet transform, which is a powerful tool to describe signals both in time (spatial) and frequency domains that has found numerous successful applications in physical and biomedical sciences.

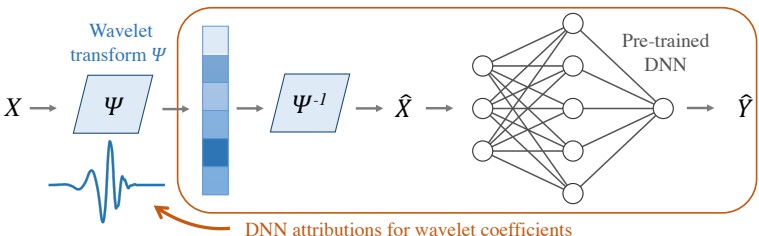

Figure 1: Adaptive wavelet distillation uses attributions from a trained DNN to improve its wavelet transform, while satisfying constraints for reconstruction error and wavelet constraints. See Eq. 8 for the precise formulation of the optimization objective.

Wavelets have many properties amenable to interpretation: they can form an orthogonal basis, identify a sparse representation of a signal, and tile different frequencies and spatial locations (and sometimes rotations), allowing for multiresolution analysis. Most previous work has focused on hand-designed wavelets for different scenarios rather than wavelets which adapt to given data. Recent work has explored wavelets which adapt to an input data distribution, under the name optimized wavelets or adaptive wavelets [11–19]. Moreover, some work has used wavelets as part of the underlying structure of a neural network, as in wavelet networks / wavelet neural networks [20–25], or the scattering transform [26, 27]. However, none of them utilize wavelets for interpretable model distillation.

Fig 1 outlines Adaptive Wavelet Distillation (AWD), our approach for distilling a wavelet transform from a trained DNN. A key novelty of AWD is that it uses *attributions from a trained DNN* to improve the learned wavelets;[2] this incorporates information not just about the input signals, as is done in previous work, but also about the target variable and the inductive biases present in the DNN.[3]

This paper deviates significantly from a typical NeurIPS paper. While there has been an explosion of work in "interpretable machine learning" [28], there has been very limited development and grounding of these methods *in the context of a particular problem and audience*. This has led to much confusion about how to develop and evaluate interpretation methods [29, 30]; in fact, a major part of the issue is that interpretability cannot be properly defined without the context of a particular problem and audience [3]. As interpretability and scientific machine learning enter a new era, researchers must ground themselves in real-world problems and work closely with domain experts.

This paper focuses on scientific machine learning—providing insight for a particular scientific audience into a chosen scientific problem— and from its outset, was designed to solve a particularly challenging cosmology problem in close collaboration with cosmologists. We showcase how AWD can inform relevant features in a fundamental problem in cosmology: inferring cosmological parameters from weak gravitational lensing convergence maps.[4] In this case, AWD identifies high-intensity peaks in the convergence maps and yields an easily interpretable model which outperforms state-of-the-art neural networks in terms of prediction performance. We next find that AWD successfully provides prediction improvements in another scientific application (now in collaboration with cell-biology experts): molecular-partner prediction. In this case, AWD allows us to vet that the model's use of clathrin corresponds to our domain knowledge about how clathrin must build up slowly then fall in order to predict a successful event. In both cases, the wavelet models from AWD concisely explains model behavior using extremely few parameters (e.g. 10), while also extracting compressed

---

[2]By attributions, we mean feature importance scores given input data and a pre-trained DNN.

[3]Though we focus on DNNs, AWD works for any black-box models for which we can attain attributions.

[4]For the purpose of this work, we work with simulated lensing maps.

representations of the input in comparison to a standard wavelet model. We hope that the depth and grounding of the scientific problems in this work can spur further interpretability research in real-world problems, where interpretability can be evaluated by and enrich domain knowledge, beyond benchmark data contexts such as MNIST [31] where the need for interpretability is less cogent.

## 2 Background on wavelet transform and TRIM

### 2.1 Wavelet transform

Wavelets are a class of functions that are localized both in the time and frequency domains. In the classical setting, each wavelet is a variation of a single wavelet $\psi$, called the *mother wavelet*. A family of discrete wavelets can be created by scaling and translating the mother wavelet in discrete increments:

$$\left\{ \psi_{j,n}(t) = \frac{1}{\sqrt{2^j}} \psi\left( \frac{t - 2^j n}{2^j} \right) \right\}_{(j,n) \in \mathbb{Z}^2}, \tag{1}$$

where each wavelet in the family $\psi_{j,n}(t)$ represents a unique scale and translation of $\psi$. With a carefully constructed wavelet $\psi$ (see Appendix A.2), the family of wavelets (1) forms an orthonormal basis of $L^2(\mathbb{R})$. Namely, any signal $x \in L^2(\mathbb{R})$ can be decomposed into

$$x = \sum_n \sum_j d_j[n] \psi_{j,n}, \tag{2}$$

where the wavelet (or detail) coefficients $d_j[n]$ at scale $2^j$ are computed by taking the inner product with the basis functions, $d_j[n] = \langle x, \psi_{j,n} \rangle = \int x(t) \psi_{j,n}(t) dt$. The decomposition (2) requires an infinite number of scalings to calculate the discrete wavelet transform. To make this decomposition computable, the *scaling function* $\phi$ is introduced so that

$$x = \sum_n a_J[n] \phi_{J,n} + \sum_n \sum_j^J d_j[n] \psi_{j,n}, \tag{3}$$

where $\phi_{J,n}(t) = 2^{-J/2} \phi(2^{-J} t - n)$ represent different translations of $\phi$ at scale $2^J$ and $a_J[n] = \langle x, \phi_{J,n} \rangle$ are the corresponding approximation coefficients. Conceptually, the $\phi_{J,n}$ form an orthogonal basis of functions that are smoother at the given scale $2^J$, and therefore can be used to decompose the smooth residuals not captured by the wavelets [32].

A fundamental property of the discrete wavelet transform is that the approximation and detail coefficients at scale $2^{j+1}$ can be computed from the approximation coefficients of the previous scale at $2^j$ [33, 34]. To see this, let us define the two discrete filters, lowpass filter $h$ and highpass filter $g$

$$h[n] = \langle \frac{1}{\sqrt{2}} \phi(t/2), \phi(t-n) \rangle \quad \text{and} \quad g[n] = \langle \frac{1}{\sqrt{2}} \psi(t/2), \phi(t-n) \rangle. \tag{4}$$

Then the following recursive relations hold between the approximation and detail coefficients at two consecutive resolutions:

$$\begin{cases} a_{j+1}[p] = \sum_n h[n - 2p] a_j[n] = a_j \star \bar{h}[2p]; \\ d_{j+1}[p] = \sum_n g[n - 2p] a_j[n] = a_j \star \bar{g}[2p], \end{cases} \tag{5}$$

where we denote $\bar{h}[n] = h[-n]$ and $\bar{g}[n] = g[-n]$. Conversely, the approximation coefficients at scale $2^j$ can be recovered from the coarser-scale approximation and detail coefficients using

$$a_j[p] = \sum_n h[p - 2n] a_{j+1}[n] + \sum_n g[p - 2n] d_{j+1}[n]. \tag{6}$$

Together, these recursive relations lead to the filter bank algorithm, the cascade of discrete convolution and downsampling, which can be efficiently implemented in time $\mathcal{O}$ (Signal length). The discrete wavelet transform can be extended to two dimensions, using a separable (row-column) implementation of 1D wavelet transform along each axis (see Appendix A.1).

## 2.2 Transformation Importance (TRIM)

The work here requires the ability to compute attributions which identify important features given input data and a trained DNN. Most work on interpreting DNNs has focused on attributing importance to features in the input space of a model, such as pixels in an image or words in a document [35–39]. Instead, here we rely on TRIM (Transformation Importance) [40], an approach which attributes importance to features in a transformed domain (here, the wavelet domain) via a straightforward model reparameterization.

Formally, let $f$ be a pre-trained model that we desire to interpret. If $\Psi : \mathcal{X} \to \mathcal{W}$ is a bijective mapping that maps an input $x$ to a new domain $w = \Psi(x) \in \mathcal{W}$, TRIM reparameterizes the model as $f' = f \circ \Psi^{-1}$, where $\Psi^{-1}$ denotes the inverse of $\Psi$. In the case that $\Psi$ is not exactly invertible, TRIM adds the residuals to the output of $\Psi^{-1}$, i.e., $f'$ is reparameterized by $f'(w) = f(\Psi^{-1}w + r)$ where $w = \Psi(x)$ and $r = x - \Psi^{-1}(w)$. If $S$ indexes a subset of features in the transformed space indicating which part of the transformed input to interpret, we then define

$$\text{TRIM}_{\Psi,f}(w_S) = attr(f'; w_S), \tag{7}$$

where $attr(; w)$ is an attribution method that is evaluated at $w$ and outputs an importance value, and where $w_S$ denotes the subvector of $w$ indexed by $S$. The choice of the attribution method $attr()$ can be any local interpretation technique (e.g. LIME [35] or Integrated Gradients (IG) [37]); here we focus mainly on the saliency map [41], which simply calculates the gradient of the model's output with respect to its transformed input to define feature attribution. We leave more complex attribution methods such as IG or ACD [38] to future work.

## 3 Adaptive wavelet distillation through interpretations

Adaptive wavelet distillation (AWD) aims to learn a wavelet transform which effectively represents the input data, as well as capture information about a model trained to predict a response using the input data. Whether or not the resulting wavelet model is (i) sufficiently interpretable and (ii) predictive depends on the context of the problem. Here, we provide two scientific data problems where wavelet models satisfy both criteria (Sec 4).

We now detail how AWD wavelets can be built upon to form an extremely simple model in various contexts (see Sec 4). We first require that the wavelet transform is invertible, allowing for reconstruction of the original data. This ensures that the transform does not lose any information in the input data. We next assure that the learned wavelet is a valid wavelet: the wavelet function $\psi$ and the corresponding scaling function $\phi$ span a sequence of subspaces satisfying the multiresolution axioms [42]. Finally, we add the distillation part of AWD. We calculate the attribution scores of a given model $f$ for each coefficient in the wavelet representation, and try to find a wavelet function $\psi$ that makes these attributions sparse. Intuitively, this ensures that the learned wavelet should find a representation which can concisely explain a model's prediction. Writing the discrete wavelet transform using the discrete filters $h$ and $g$ (see Eq. 4), we now give a final optimization problem for AWD:

$$\underset{h,g}{\text{minimize}} \, \mathcal{L}(h,g) = \underbrace{\frac{1}{m}\sum_i \|x_i - \widehat{x}_i\|_2^2}_{\text{Reconstruction loss}} + \underbrace{\frac{1}{m}\sum_i W(h,g,x_i;\lambda)}_{\text{Wavelet loss}} + \underbrace{\gamma \sum_i \|\text{TRIM}_{\Psi,f}(\Psi x_i)\|_1}_{\text{Interpretation loss}}, \tag{8}$$

where $\Psi$ denotes a wavelet transform operator induced by $\psi$, and $\widehat{x}_i$ denotes the reconstruction of the data point $x_i$. Here $\lambda, \gamma > 0$ represent hyperparameters that are tuned by users. The only parameters optimized are the lowpass filter $h$ and the highpass filter $g$. The corresponding scaling and wavelet functions can be obtained from $(h,g)$ via the following mapping [32]: $\widehat{\phi}(w) = \prod_{p=1}^{\infty} \frac{\widehat{h}(2^{-p}w)}{\sqrt{2}}$ and $\widehat{\psi}(w) = \frac{1}{\sqrt{2}}\widehat{g}(w/2)\widehat{\phi}(w/2)$, where $\widehat{\phi}$ and $\widehat{\psi}$ represent the Fourier transforms of $\phi$ and $\psi$ respectively.

**Wavelet loss**  The wavelet loss ensures that the learned filters yield a valid wavelet transform. In contrast to the wavelet constraints used in [11], our formulation introduces additional terms that ensure almost sufficient and necessary conditions on the filters $(h,g)$ to build an orthogonal wavelet basis. Specifically, [32, Theorem 7.2] states the following sufficient conditions on the lowpass filter:

if $h$ satisfies

$$\sum_n h[n] = \sqrt{2} \quad \text{and} \quad |\widehat{h}(w)|^2 + |\widehat{h}(w+\pi)|^2 = 2 \text{ for all } w, \tag{9}$$

as well as some mild conditions, it can generate a scaling function such that the scaled and translated family of the scaling function forms an orthonormal basis of the space of multiresolution approximations of $L^2(\mathbb{R})$. [43, Theorem 3] further shows that the orthogonality of translates of the scaling function implies that the lowpass filter is orthogonal after translates by 2, i.e.,

$$\sum_n h[n]h[n-2k] = \begin{cases} 1 & \text{if } k=0 \\ 0 & \text{otherwise} \end{cases}, \quad \text{and as a result, } \|h\|_2 = 1. \tag{10}$$

Hence the conditions (9), (10) characterize the almost sufficient and necessary conditions on the lowpass filter. Moreoever, [32, Theorem 7.3] shows that the valid highpass filter can be constructed from the lowpass filter: in the time domain, it can be written as

$$g[n] = (-1)^n h[N-1-n], \tag{11}$$

where $N$ is the support size of $h$. Together with (10), we can also deduce that the highpass filter has mean zero, i.e., $\sum_n g[n] = 0$ which is necessary for the filter $g$. See Appendix A.2 for further details.

Finally, we want the learned wavelet to provide sparse representations so we add the $\ell_1$ norm penalty on the wavelet coefficients. Combining all these constraints via regularization terms, we define the wavelet loss at the data point $x_i$ as

$$W(h, g, x_i; \lambda) = \lambda \|\Psi x_i\|_1 + (\sum_n h[n] - \sqrt{2})^2 + (\sum_n g[n])^2 + (\|h\|_2^2 - 1)^2$$
$$+ \sum_w (|\widehat{h}(w)|^2 + |\widehat{h}(w+\pi)|^2 - 2)^2 + \sum_k (\sum_n h[n]h[n-2k] - \mathbf{1}_{k=0})^2,$$

where $g$ is set as in (11) and $\lambda > 0$ controls strength of the sparsity of the wavelet representations. We enforce the penalty $(|\widehat{h}(w)|^2 + |\widehat{h}(w+\pi)|^2 - 2)^2$, only at the discrete values of $w \in \{\frac{2\pi k}{N}, k = 1, \dots, N\}$ through the discrete Fourier transform. Notice that the wavelet loss does not introduce any additional hyperparameters besides $\lambda$. In fact, we empirically observe that the sum of penalty terms, except the sparsity penalty, remains very close to zero as long as the filters $(h, g)$ are initialized using known wavelet filters and the interpretation loss is not enforced too strongly.

**Interpretation loss** The interpretation loss enables the distillation of knowledge from the pre-trained model $f$ into the wavelet model. It ensures that attributions in the space of wavelet coefficients $\Psi x_i$ are sparse, where the attributions of wavelet coefficients is calculated by TRIM [40], as described in Sec 2.2. This forces the wavelet transform to produce representations that concisely explain the model's predictions at different scales and locations. The hyperparameter $\gamma$ controls the overall contribution of the interpretation loss; large values of $\gamma$ can result in large numerical differences from satisfying the conditions of the mathematical wavelet filters. To our knowledge, this is the first method which uses interpretations from a pre-trained model to improve a wavelet representation. This enables the wavelets to not only adapt to the distribution of the inputs, but also gain information about the predicted outputs through the lens of the model $f$.

## 4  AWD improves interpretability, prediction performance, and compression in two scientific problems and in simulations

Fig 2 shows a visual schematic of the distillation and prediction setup for one synthetic and two scientific data problems in this section, whose details will be discussed in the following subsections.[5] In both scientific problems, we build extremely simple models based on AWD which significantly outperform the state-of-the-art DNN performance with far fewer number of parameters.[6]

---

[5] In all experiments, the wavelet function is computed from the corresponding lowpass filter using the *PyWavelets* package [44] and building on the *Pytorch Wavelets* [45, Chapter 3] package.

[6] For example, the final molecular-partner prediction model contains only 10 parameters for the low-pass filter, along with only 30 coefficients in the sparse linear model. The final distilled cosmology model learns only 10 parameters for the low-pass filter to make its predictions.

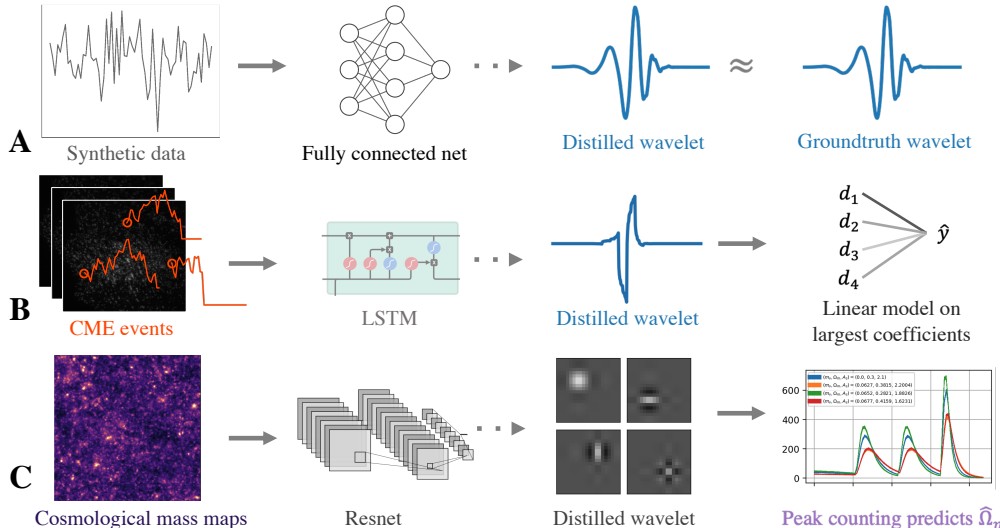

Figure 2: Distillation and prediction setup for the three scenarios in Sec 4. (**A**) In synthetic simulations, AWD is able to recover groundtruth wavelet (DB5) that are linked to a response variable (Sec 4.1). (**B**) Wavelets distilled by AWD from an LSTM trained to predict molecular partners capture biologically meaningful properties of a large build up in clathrin fluorescence followed by a sharp drop and enable prediction using only a few key coefficients (Sec 4.2). (**C**) AWD finds wavelets that are efficient at capturing cosmological information in weak lensing convergence maps and can improve state-of-the-art performance of cosmological parameter inference using an AWD-based simple peak-counting algorithm (Sec 4.3). Note that the model forms in (**B**) and (**C**) come from knowledge about the domain problem.

## 4.1 Synthetic data

We begin our evaluation using simulations to verify whether AWD can recover groundtruth wavelets from noisy data. In these simulations, the inputs $x_i$ are generated i.i.d. from a standard Gaussian distribution $\mathcal{N}(0, 1)$. To generate the response variable, the inputs are transformed into the wavelet domain using Daubechies (DB) 5 wavelets [46], and the response is generated from a linear model $y_i = \langle \Psi x_i, \beta \rangle + \epsilon_i$, where the true regression coefficients are 2 for a few selected locations at a particular scale and 0 otherwise; the noise $\epsilon_i$ is generated i.i.d. from a Gaussian distribution $\mathcal{N}(0, 0.1^2)$. Then, a 3-layer fully connected neural network with ReLU activations is trained on the pairs of $x_i, y_i$ to accurately predict this response. Note that for any non-singular matrix $A$, the mapping $x \mapsto \langle A^{-1}\Psi x, A^\top \beta \rangle$ predicts the response equally well, but the representations in the groundtruth wavelet explain the model's prediction most concisely. The challenge is then to accurately distill the groundtruth wavelet (DB5) from this DNN. This task is fairly difficult: AWD must not only select which scale and locations are important, it must also precisely match the shape of $h$ and $g$ to the groundtruth wavelet.

Fig 3 shows the performance of AWD on this task. We initialize the AWD lowpass filter to different known lowpass filters corresponding to DB5+random noise (support size 10), Symlet 5 (support size 10), and Coiflet 2 (support size 12), as shown in Fig 3(**A**), and then minimize the objective in Eq. 8. In order to recover the groundtruth, we select hyperparameters $\lambda$ and $\gamma$ that minimize the distance to the groundtruth wavelet $\psi^\star$. Distance is measured via $d(\psi, \psi^\star) = \min\{\min_k \|\psi^k - \psi^\star\|_2, \min_k \|\widetilde{\psi^k} - \psi^\star\|_2\}$, where $\psi^k$ is the wavelet $\psi$ circular shifted by $k$ and $\widetilde{\psi}$ is the wavelet $\psi$ flipped in the left/right direction. That is, d calculates the minimum $\ell_2$ distance between two wavelets under circular shifts and left/right flip. When the two wavelets have different size of support, the shorter wavelet is zero-padded to the length of the longer [11]. Fig 3(**B**) shows that for each different initialization, we find that the distilled wavelet gets considerably closer to the groundtruth wavelet. In particular, the results for DB5+noise and Coiflet 2 are nearly identical to the groundtruth and cannot be distinguished in the plot. This is particularly impressive since the

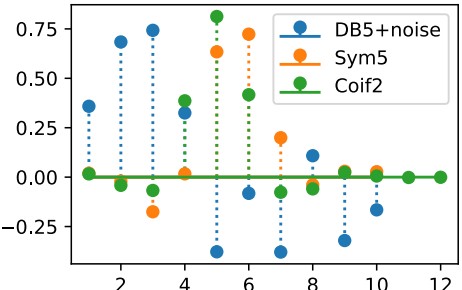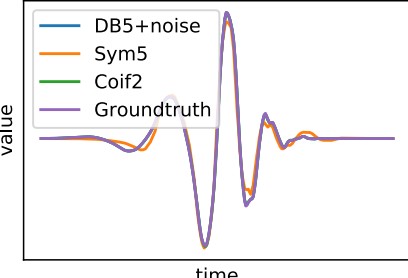

Figure 3: AWD accurately identifies the groundtruth important wavelet in simulated data. (**A**) Plots of the initial lowpass filters. (**B**) Final wavelets extracted by AWD.

support size of Coiflet 2 differs from that of the groundtruth wavelet, making the task more difficult. Overall, these results demonstrate the ability of AWD to distill key information out of a pre-trained neural network.

## 4.2 Molecular partner-prediction for a central process in cell biology

We now turn our attention to a crucial question in cell biology related to the internalization of macromolecules via clathrin-mediated endocytosis (CME) [47]. CME is the primary pathway for entry into the cell, making it essential to eukaryotic life [48]. CME is an orchestra consisting of hundreds of different protein dynamics, prompting a line of research aiming to better understand this process [49]. Crucial to understanding CME is the ability to readily distinguish whether or not the recruitment of certain molecules will allow for endocytosis, i.e., successfully transporting an object into a cell. Previous approaches have largely relied on the presence of a specific scission/uncoating marker during imaging [50, 51]. Alternatively, previous works use domain knowledge to hand-engineer features based on the lifetime of an event or thresholds on the recruited amplitude of the clathrin molecule [52, 53].

Here, we aim to identify successful CME events with a learning approach, obviating the need for an auxiliary marker or hand-engineered features. We use a recently published dataset [50] which tags two molecules: clathrin light chain A, which is used as the predictor variable, and auxilin 1, the target variable. In this context, clathrin is used to track the progress of an event, (as recruitment of clathrin molecules usually precedes scission) and recruitment of auxilin molecules follows only when endocytosis successfully occurs (to facilitate disassembly of the clathrin-coated vesicle). See data details in Appendix C. Time-series of fluorescence amplitudes (see Fig 2B) are extracted from raw cell videos for clathrin [52] and used to predict the mean amplitude of the auxilin signal, an indicator of whether an event was successful or not. The dataset is randomly split into a training set of 2,936 data units of dimension 40 and a test set of 1,005 data units. This is a challenging problem where deep learning has recently been shown to outperform classical methods. We train a DNN (an LSTM [54]) to predict the auxilin response from the clathrin signal. The model shows state-of-the-art prediction performance, but has extremely poor interpretability and computational cost, so we aim here to distill it into a wavelet model through AWD.

Fig 4 shows qualitatively how the learned wavelet function $\psi$ changes as a function of the interpretation penalty $\gamma$ (increasing to the right) and the sparsity penalty $\lambda$ (increasing downwards). In the initial stage of training, we initialize the lowpass filter to correspond to the Daubechies (DB) 5 wavelet. Different combinations of the penalties lead to vastly different learned wavelets, though they all tend to reveal edge-detecting characteristics for a reasonable range of hyperparameter values.

We now test the distilled wavelets for their predictive power. To create an extremely transparent model, we extract only the maximum 6 wavelet coefficients at each scale. By taking the maximum coefficients, these features are expected to be invariant to the specific locations where a CME event occurs in the input data. This results in a final model with 30 coefficients (6 wavelet coefficients at 5 scales). These wavelet coefficients are used to train a linear model, and the best hyperparameters are selected via cross-validation on the training set. Fig 2 shows the best learned wavelet (for one particular run) extracted by AWD corresponding to the setting of hyperparameters $\lambda = 0.005$

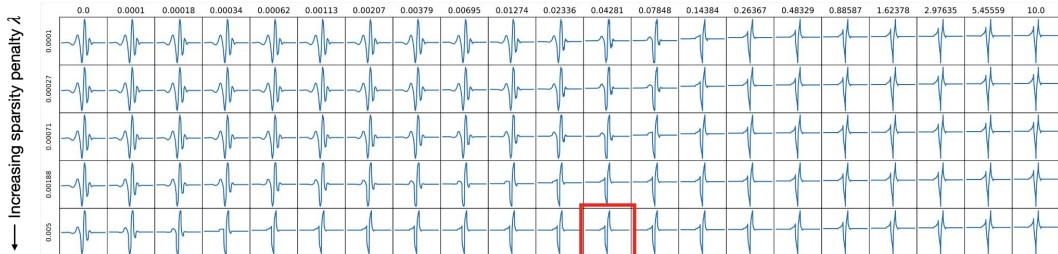

Figure 4: Varying sparsity and interpretation penalty yields different valid wavelets. Wavelet highlighted in red is selected by cross-validation and yields the best prediction performance.

and $\gamma = 0.043$. Table 1 compares the results for AWD to the original LSTM and the initialized, non-adaptive DB5 wavelet model, where the performance is measured via a standard $R^2$ score, a proportion of variance in the response that is explained by the model. The AWD model not only closes the gap between the standard wavelet model (DB5) and the neural network, it considerably improves the LSTM's performance (a 10% increase of $R^2$ score). Table 1 also includes predictive performance of the adaptive wavelet when the interpretation loss is removed during the training; it still outperforms the baseline wavelet (DB5), but fails to outperform LSTM. Moreover, we calculate how well the AWD wavelet compresses its input. Specifically, we measure the proportion of wavelet coefficients in the test set in which the magnitude and the attributions are both above $10^{-3}$. The AWD wavelet exhibits much better compression than DB5 (an 18% reduction), showing the ability of AWD to simultaneously provide sparse representations and explain the LSTM's predictions concisely. The AWD model also dramatically decreases the computation time at test time, a more than 200-fold reudction when compared to LSTM.

In addition to improving prediction accuracy, AWD enables domain experts to vet their experimental pipelines by making them more transparent. By inspecting the learned wavelet, AWD allows for checking what clathrin signatures signal a successful CME event; it indicates that the distilled wavelet aims to identify a large buildup in clathrin fluorescence (corresponding to the building of a clathrin-coated pit) followed by a sharp drop in clathrin fluorescence (corresponding to the rapid deconstruction of the pit). This domain knowledge is extracted from the pre-trained LSTM model by AWD using only the saliency interpretations in the wavelet space.

Table 1: Performance comparisons for different models in molecular-partner prediction. AWD substantially improves predictive accuracy, compression, and computation time on the test set. A higher $R^2$ score, and lower compression factor, and lower computation time indicate better results. For AWD, values are averaged over 5 different random seeds.

|  | **AWD** | Standard Wavelet (DB5) | LSTM | AWD w/o interp. loss |
|---|---|---|---|---|
| Regression ($R^2$ score) | **0.262 (0.001)** | 0.197 | 0.237 | 0.231 (0.001) |
| Compression factor | **0.574 (0.010)** | 0.704 | N/A | 0.651 (0.003) |
| Computation time | **0.0002s** | 0.0002s | 0.0449s | 0.0002s |

### 4.3 Estimating a fundamental parameter surrounding the origin of the universe

We now focus on a cosmology problem, where AWD helps replace DNNs with a more interpretable alternative. Specifically, we consider weak gravitational lensing convergence maps, i.e., maps of the mass distribution in the universe integrated up to a certain distance from the observer. In a cosmological experiment (e.g. a galaxy survey), these mass maps are obtained by measuring the distortion of distant galaxies caused by the deflection of light by the mass between the galaxy and the observer [55]. These maps contain a wealth of physical information of interest, such as the total matter density in the universe, $\Omega_m$. Current cosmology research aims to identify the most informative features in these maps for inferring the cosmological parameters such as $\Omega_m$. The traditional summary statistic for lensing maps is the power spectrum which is known to be sub-optimal for parameter

inference. Tighter parameter constraints can be obtained by including higher-order statistics, such as the bispectrum [56] and peak counts [57]. However, DNN-based inference methods claim to improve on constraints based on these traditional summaries [58–60].

Here, we aim to improve the predictive power of DNN-based methods while gaining interpretability by distilling a predictive AWD model. In this context, it is critically important to obtain interpretability, as it provides deeper understanding into what information is most important to infer $\Omega_m$ and can be used to design optimal experiments or analysis methods. Moreover, because these models are trained on numerical simulations (realizations of the Universe with different cosmological parameters), it is important to validate that the model uses reasonable features rather than latching on to numerical artifacts in the simulations. We start by training a model to predict $\Omega_m$ from simulated weak gravitational lensing convergence maps. We train a DNN[7] to predict $\Omega_m$ from 100,000 mass maps simulated with 10 different sets of cosmological parameter values at the universe origin from the `MassiveNuS` simulations [62] (full simulation details given in Appendix D), achieving an $R^2$ value of 0.92 on the test set (10,000 mass maps); Fig 2C shows an example mass map.

We again construct an interpretable model using the wavelets distilled by AWD from the trained DNN. To make predictions, we use the simple peak-counting algorithm developed in a previous work [59], which convolves a peak-finding filter with the input images. Then, these peaks are used to regress on the outcome. In contrast to the fixed filters such as Laplace or Roberts cross used in previous works [59], here we use the wavelets distilled by AWD, which result in three 2D wavelet filters (LL, LH, HL) and the 2D approximation filter (LL). The size of the distilled AWD filters is 12×12 and inspection of these filters shows a majority of the mass is concentrated on 3×3 subfilters (see Fig 2C). Then we extract those subfilters to use for peak-finding filters—by doing so, the size of the filters match with those used in [59] (additional details given in Appendix D.1). The hyperparameters for AWD are selected by evaluating the predictive model's performance on a held-out validation set.

Table 2 shows the results of predicting using the peak-finding algorithm with various filters. The evaluation metric is the RMSE (Root mean square error). Its performance again outperforms the fully trained neural network (Resnet) model and the standard non-adaptive wavelet (DB5) model, as well as other baseline methods using Laplace filter and Roberts cross filter (see Appendix D.1 for details on how these filters are defined). Without the interpretation loss, the adaptive wavelet fails to outperform the Roberts cross filter and Resnet model. Moreover, as can be seen in the compression factor, the AWD wavelet provides more efficient representations for the mass maps as well as concise explanation for the DNN's predictions compared to the DB 5 wavelet.

Table 2: Performance comparisons for different models in cosmological parameter prediction. The lower RMSE and compression factor indicate better results. For RMSE, standard deviations are estimated from $10,000$ bootstrap samples.

| | **AWD** | Roberts-Cross | Laplace | DB5 Wavelet | Resnet | AWD w/o interp. loss |
|---|---|---|---|---|---|---|
| Regression (RMSE $\times 10^{-2}$) | **1.029 (0.033)** | 1.259 (0.039) | 1.369 (0.047) | 1.569 (0.048) | 1.156 (0.024) | 1.354 (0.047) |
| Compression factor | **0.610** | N/A | N/A | 0.620 | N/A | 0.616 |

Fig 2C shows the learned AWD filters corresponding to the best distilled wavelet. The learned wavelet filters are symmetric and resemble the "matched filters" which have been used in the past to identify peaks on convergence maps in the cosmology literature [63, 64]. We expect from cosmology knowledge that much information is contained in the peaks of the convergence maps (their amplitude, shape, and numbers), so this indeed matches our expectations based on physics. The high predictive performance further demonstrates that the AWD filters are more efficient at capturing cosmological information and better adapted to the shape of the peaks, than standard wavelets could do.

Moreover, the adaptive wavelet distillation allows us to look at "wavelet activation maps" (see Fig D2) to localize where in the convergence maps important information is concentrated. In other words,

---

[7]The model's architecture is Resnet 18 [61], modified to take only one input channel.

we can indeed see that the AWD wavelet concentrates on identifying high intensity peaks, which is where most of the "localized" information is expected from theory.

## 5   Discussion

In this work, we introduce AWD, a method to distill adapative wavelets from a pre-trained supervised model such as DNNs for interpretation. Doing so enables AWD to automatically detect and adapt to aspects of data that are important for prediction in an interpretable manner. The benefits of distilling relevant predictive information captured in a DNN are demonstrated through applications to synthetic and real data in two scientific settings. Overall, AWD allows us to interpret a DNN in terms of conventional wavelets, bringing interpretability with domain insights while simultaneously improving compression and computational costs, all while preserving or improving predictive power.

**Limitations and future work**    AWD works well in particular domains where wavelets are a reasonable modeling choice, such as images and time-series, which possess multi-scale structure. Other domains, such as DNA-sequence data or text data which do not posses this structure would not benefit from AWD. Therefore, it is important to use domain knowledge to decide whether AWD can be beneficial for a given task and domain. We also used domain knowledge to pick reasonable choices for the number of scales in the distilled wavelet transform, and achieved strong performance without tuning this parameter. Future work could include more analysis on the effect of the number of scales, or an optimization procedure to pick it.

In this paper we test our method with the saliency attribution method; however, many alternative interpretation techniques have been proposed recently, such as SHAP [36] or Contextual Decomposition [38, 65], and the comparison between different interpretation techniques can be carefully explored in the context of a particular problem and audience. Care must be taken to ensure that these methods can be efficiently optimized, as even the simplest saliency methods requires taking a second partial derivative during distillation, which is computationally expensive especially for large data and network sizes.

Throughout the experiments, the learned wavelets are initialized to a filter corresponding to a known wavelet. Initializing to a known wavelet filter makes the optimization faster and more stable, but when initializing randomly, the predictive performance can be unstable, especially when the variance of the initialization is high. However, this can be solved by initializing at any known wavelet (potentially with some random noise added to it) or first performing a few steps of the optimization without the interpretation loss before adding it in. Another promising strategy is initializing with an (unsupervised) dictionary of wavelets [19].

The current work learns a single-layer wavelet transform, but the complex nature of modern datasets often require strong nonlinearities. Future work could extend AWD beyond a single-layer wavelet transform, e.g. by borrowing ideas from scattering transform [27] or to other interpretable models [2, 66]. This would allow for bridging closer to deep learning while keeping interpretability, which can be effectively applied to other areas, such as computer vision and natural-image classification. We hope to continue this line of research in order to improve the interpretability and computational efficiency of DNN models across many domains ranging from physical and biomedical sciences to computer vision and information technology.

**Societal impacts**    AWD generally helps alleviate many concerns around DNNs, both in terms of interpretability and computational resources. However, its optimization at training time necessitates training a DNN, which can incur high computational cost (and therefore corresponding impacts from energy usage). Additionally, the interpretability granted by AWD could lead to over-reliance on the fitted models, even in applications where it does not perform well.

## Acknowledgements

We gratefully acknowledge partial support from NSF TRIPODS Grant 1740855, DMS-1613002, 1953191, 2015341, IIS 1741340, ONR grant N00014-17-1-2176. Moreover, this work is supported in part by the Center for Science of Information (CSoI), an NSF Science and Technology Center, under grant agreement CCF-0939370, by the NSF Grant DMS 2031883 "Collaboration on the Theoretical

Foundations of Deep Learning", and by the NSF Foundations of Data Science Institute (FODSI). SU was supported with funding from Philomathia Foundation and Chan Zuckerberg Initiative Imaging Scientist program. The authors would also like to thank Alan Dong for enlightening discussions on optimizing wavelets. The authors would also like to thank Tom Kirchhausen, Kangmin He, Eli Song, and Song Dang for providing the clathrin mediated endocytosis data to apply AWD for molecular partner predictions. We would like to gratefully acknowledge AWS computing credits as part of NSF Award 1741340.

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
