# Appendix

## A   Further wavelet details

### A.1   2D wavelet transform

The discrete wavelet transform can be extended to two dimensions, using a separable (row-column) implementation of 1D wavelet transform along each axis. In 2D, the family of wavelets is characterized by the following three wavelets

$$\psi^1(x_1, x_2) = \phi(x_1)\psi(x_2),\ \psi^2(x_1, x_2) = \psi(x_1)\phi(x_2),\ \psi^3(x_1, x_2) = \psi(x_1)\psi(x_2),$$

named LH, HL, HH wavelets respectively. Together with the scaling function $\widetilde{\phi}(x) = \phi(x_1)\phi(x_2)$, the 2D discrete wavelet transform gives four components at each iteration, contrary to the 1D case, by applying the decomposition formula (5) to the separable wavelets and scaling functions

$$\begin{cases} a_{j+1}[p] = a_j \star \bar{h}\bar{h}[2p]; \\ d_{j+1}^1[p] = a_j \star \bar{h}\bar{g}[2p]; \\ d_{j+1}^2[p] = a_j \star \bar{g}\bar{h}[2p]; \\ d_{j+1}^3[p] = a_j \star \bar{g}\bar{g}[2p], \end{cases}$$

for $p = (p_1, p_2)$, where for 2D discrete filters we denote $hh = h[n_1]h[n_2]$. In particular, the decomposition yields three detail coefficients where the highpass filter $h$ is applied to either of the two-dimensional directions or both. These coefficients are intended to represent the signal in different orientations, i.e., vertical, horizontal, and diagonal. Similarly to (6), the approximation coefficient $a_j$ at scale $2^j$ can also be recovered from the approximation coefficient $a_{j+1}$ and detail coefficients $d_{j+1}^k$, $k = 1, 2, 3$, at scale $2^{j+1}$ with formula

$$a_j[p] = [a_{j+1}]_{\uparrow 2} \star hh[p] + [d_{j+1}^1]_{\uparrow 2} \star hg[p] + [d_{j+1}^2]_{\uparrow 2} \star gh[p] + [d_{j+1}^3]_{\uparrow 2} \star g[p],$$

where $[a]_{\uparrow 2}$ denotes upsampling of the image $a$ by a factor 2.

### A.2   Conditions for orthonormal wavelet basis

This section provides further details on constructing a valid wavelet $\psi$ such that the family $\{\psi_{j,n}\}_{(j,n) \in \mathbb{Z}^2}$ of wavelets forms an orthonormal basis of $L^2(\mathbb{R})$. To do so, we introduce multiresolution analysis [33, 34] which constructs an orthonormal wavelet basis through approximations of signals at various resolutions. The key idea is that one builds a sequence of approximations for a signal with increasing resolutions while the difference between two consecutive approximations can be captured by the wavelet decomposition at a given scale.

To begin with, let $\phi$ be a scaling function in $L^2(\mathbb{R})$. To motivate the idea of multiresolution analysis, we assume that $\phi$ is the Haar scaling function, defined as

$$\phi(t) = \begin{cases} 1 & \text{if } 0 \leq t < 1 \\ 0 & \text{otherwise} \end{cases}.$$

Let $V_j$ denote the space spanned by $\{\phi_{j,n}\}_{n \in \mathbb{Z}}$, where $\phi_{j,n}(t) = 2^{-j/2}\phi(2^{-j}t - n)$. Then $V_j$ is the set of piecewise constant functions over $[2^j n, 2^j(n+1))$ for $n \in \mathbb{Z}$. The approximations of a signal $x$ at scale $2^j$ is defined by the orthogonal projection of $x$ on $V_j$, which is the closest piecewise constant function on intervals of size $2^j$. For two consecutive approximation spaces $V_j$ and $V_{j+1}$, the relation $V_{j+1} \subset V_j$ holds because any function that is constant over $[2^{j+1}n, 2^{j+1}(n+1))$ is also constant over $[2^j n, 2^j(n+1))$. Moreover, it is easy to see that $\lim_{j \to \infty} V_j = \{0\}$ and $\lim_{j \to -\infty} V_j = L^2(\mathbb{R})$.

More generally, the sequence $\{V_j\}_{j \in \mathbb{Z}}$ of subspaces with $\{0\} \subset \ldots \subset V_1 \subset V_0 \subset V_{-1} \subset \ldots \subset L^2(\mathbb{R})$ is called a multiresolution approximation if it satisfies certain properties (see [32, Definition 7.1]). The piecewise constant approximations induced by the Haar scaling function is a special case that verifies the properties of a multiresolution approximation. The multiresolution approximation is entirely characterized by the scaling function $\phi$ since the family $\{\phi_{j,n}\}_{n \in \mathbb{Z}}$ forms an orthonormal basis of $V_j$ for all $j \in \mathbb{Z}$. Remarkably, the following theorem due to [33, 34] further shows that a scaling function can be entirely determined by a discrete filter $h$ that is defined on the set of discrete values:

**Theorem 1** (Theorem 7.2 [32]). *For a discrete filter $h[n]$, if the Fourier series $\widehat{h}(w)$ is $2\pi$ periodic and continuously differentiable in a neighborhood of $w = 0$, if it satisfies*

$$\widehat{h}(0) = \sum_n h[n] = \sqrt{2} \ \text{ and } \ |\widehat{h}(w)|^2 + |\widehat{h}(w+\pi)|^2 = 2 \ \text{ for all } w,$$

*and if $\inf_{w \in [-\pi/2, \pi/2]} |\widehat{h}(w)| > 0$, then*

$$\widehat{\phi}(w) = \prod_{p=1}^{\infty} \frac{\widehat{h}(2^{-p}w)}{\sqrt{2}},$$

*is the Fourier transform of a scaling function $\phi \in L^2(\mathbb{R})$. Namely, the sequence $\{V_j\}_{j \in \mathbb{Z}}$ of subspaces induced by $\phi$ satisfies the properties of a multiresolution approximation.*

Moreover, it can be shown that any scaling function $\phi$ determines the lowpass filter $h$ via $h[n] = \langle \frac{1}{\sqrt{2}} \phi(t/2), \phi(t-n) \rangle$ (see Eq. 4). Hence Theorem 1 provides equivalence between the scaling function and the discrete lowpass filter. The next theorem states a necessary condition on the lowpass filter:

**Theorem 2** (Theorem 3 [43]). *If $\phi$ is a valid scaling function, then*

$$\sum_n h[n]h[n-2k] = \begin{cases} 1 & \text{if } k = 0 \\ 0 & \text{otherwise} \end{cases}.$$

Theorem 1 and Theorem 2 characterize the sufficient and necessary conditions on the lowpass filter to build a valid scaling function.

Next, the multiresolution approximation requires $V_j \subset V_{j-1}$ for all $j \in \mathbb{Z}$ and the details that appear at the scale $2^{j-1}$ but disappear at the coarser scale $2^j$ can be characterized by the wavelet coefficients. Indeed, if $W_j$ denotes the orthogonal complement of $V_j$ in $V_{j-1}$, i.e., $V_{j-1} = V_j \oplus W_j$, one can construct a family of wavelets $\{\psi_{j,n}\}_{n \in \mathbb{Z}}$ that forms an orthonormal basis of $W_j$:

**Theorem 3** (Theorem 7.3 [32]). *Let $\phi$ be a scaling function and $h$ the corresponding filter. Let $\psi$ be the function having a Fourier transform*

$$\widehat{\psi}(w) = \frac{1}{\sqrt{2}} \widehat{g}\left(\frac{w}{2}\right) \widehat{\phi}\left(\frac{w}{2}\right),$$

*with*

$$\widehat{g}(w) = e^{-iw}\widehat{h}^*(w+\pi).$$

*Then for any scale $2^j$, $\{\psi_{j,n}\}_{n \in \mathbb{Z}}$ is an orthonormal basis of $W_j$ and for all scales, $\{\psi_{j,n}\}_{(j,n) \in \mathbb{Z}^2}$ is an orthonormal basis of $L^2(\mathbb{R})$.*

In the time domain, the equation $\widehat{g}(w) = e^{-iw}\widehat{h}^*(w+\pi)$ can be converted to

$$g[n] = (-1)^n h[N-1-n], \tag{12}$$

where $N$ is the support size of $h$. Moreover, it follows from $\sum_n h[n] = \sqrt{2}$ (Theorem 1) and $\sum_n h[n]h[n-2k] = \mathbf{1}_{k=0}$ (Theorem 2) that $\sum_n h[2n] = \sum_n h[2n+1]$ holds [43, Theorem 2]. Using this identity, it is easy to check that the highpass filter must have zero-mean, i.e.,

$$\sum_n g[n] = 0. \tag{13}$$

Then Eq. 12 and Eq. 13 provides the sufficient and necessary conditions on the highpass filter to build a valid wavelet $\psi$.

## B  Synthetic data details

In this section, we show additional results for the experiments with synthetic data in Sec 4.1. For this task, we generate data from a linear model $y_i = \langle \Psi x_i, \beta \rangle + \epsilon_i, i = 1, \ldots, n$, where:

- The inputs $x_i \in \mathbb{R}^{n \times d}$ are generated with i.i.d. $\mathcal{N}(0,1)$ entries, where the number of input features is $d = 64$;

- $\Psi$ is given a wavelet transform operator with DB 5 wavelets;

- The noise $\epsilon_i \in \mathbb{R}^n$ is generated with i.i.d. $\mathcal{N}(0, 0.1^2)$ entries;

- The true coefficient $\beta$ is given $\beta_i = 2$ for 3 selected locations at a particular scale, and $\beta_i = 0$ otherwise.

The data is randomly split into a training set of $50,000$ data points and a test set of $5,000$ data points. Then a 3-layer fully connected neural network with 32 hidden neurons each is trained on the training set with a learning rate of $0.01$ for 20 epochs, achieving an $R^2$ score $> 0.99$ on the test set.

To distill the groundtruth wavelet (DB5) from this DNN, we solve the minimization problem given in Eq. 8 for varying hyperparameters. Here we use a warm start strategy in which we solve the problem Eq. 8 for one pair of values for hyperparameters $\lambda$ and $\gamma$ and use this solution to initialize the AWD filter at the next values of hyperparameters. In the initial stage of training, the AWD filter is initialized to the known lowpass filters corresponding to the DB 5 wavelet, Sym 5 wavelet, and Coif 2 wavelet, respectively (for DB 5, we add a noise to the lowpass filter). For each pair of the hyperparameters, the AWD filters were trained for 50 epochs with Adam optimizer with a learning rate of $0.001$. All experiments were run on an AWS instance of p3.16xlarge for a few days.

## B.1 Additional results on synthetic data

Here we show the learned wavelets as the interpretation penalty $\gamma$ and the sparsity penalty $\lambda$ vary across a sequential grid of values spaced evenly on a log scale. Fig B1 shows the results when the AWD filter in the initial stage is initialized to the lowpass filter corresponding to the DB 5 wavelet + noise; Fig B2 shows the results when the AWD filter in the initial stage is initialized to the lowpass filter corresponding to the Sym 5 wavelet; and Fig B3 shows the results when the AWD filter in the initial stage is initialized to the lowpass filter corresponding to the Coif 2 wavelet. We can see that as long as the interpretation penalty is not too small or large, the wavelets distilled by AWD accurately recovers the groundtruth (DB 5) wavelet.

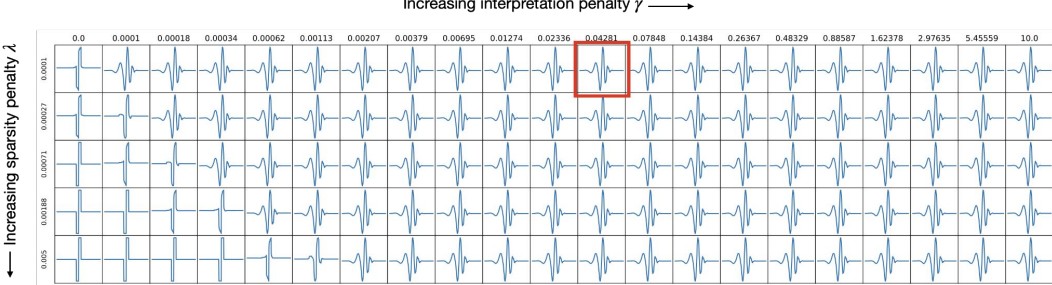

Figure B1: Varying sparsity and interpretation penalty yields different valid wavelets. In the initial stage, the AWD filter is initialized to the lowpass filter corresponding to DB 5 + noise.

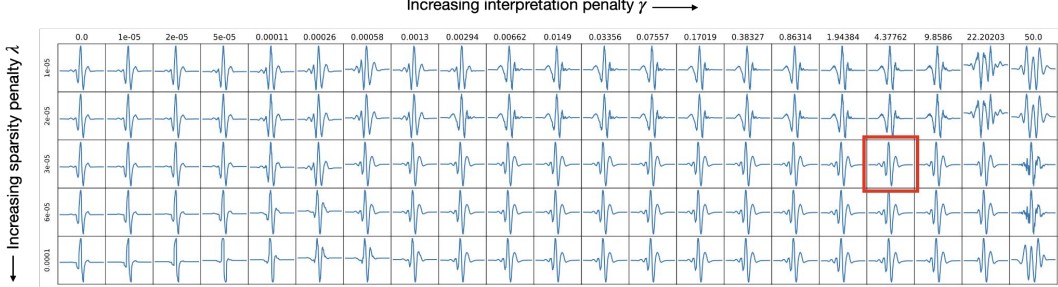

Figure B2: Varying sparsity and interpretation penalty yields different valid wavelets. In the initial stage, the AWD filter is initialized to the lowpass filter corresponding to Sym 5.

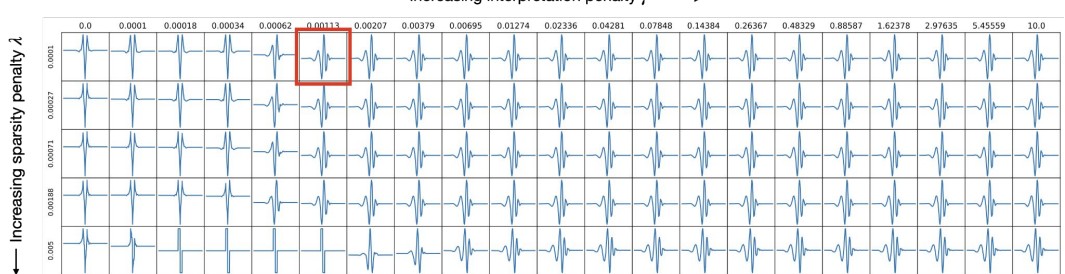

Figure B3: Varying sparsity and interpretation penalty yields different valid wavelets. In the initial stage, the AWD filter is initialized to the lowpass filter corresponding to Coif 2.

Fig B4 calculates the distance between the learned wavelets and the groundtruth (DB5) wavelet, defined as in Sec 4.1, as the interpretation penalty varies. When initialized at DB 5+noise, the learned wavelets get very close to the groundtruth wavelet for a wide range of $\gamma$ values, regardless of different sparsity penalty. On the other hand, when initialized at Sym 5, AWD can accurately recover the groundtruth wavelet only at the large values of $\lambda$; whereas for Coif 2, AWD can recover the groundtruth wavelet only at the small values of $\lambda$.

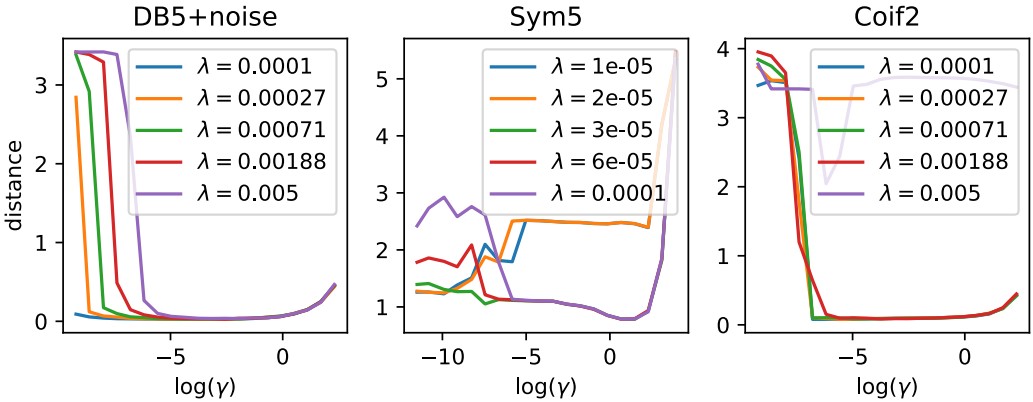

Figure B4: The distance between the learned wavelets and the groundtruth wavelet, defined as in Sec 4.1, is plotted against $\log(\gamma)$ for different values of $\lambda$.

## C    Molecular partner-prediction details

This section gives an overview of the preprocessing for the clathrin-mediated endocytosis problem in Appendix C.2. For a detailed overview of the data, see the original study [50]. In order to convert the raw fluorescence images to time-series traces, we use tracking code from previous work [52]. The tracking fits a Gaussian curve to the images (with standard deviation given by the imaging parameters). When the fit to the first channel (i.e. clathrin) is significant,[8] the track is recorded and a fit is forced to the second channel (i.e. auxilin). The amplitudes of each track over time are then extracted. Fig C1 shows some examples of extracted clathrin traces.

The architecture of the LSTM used in this work has one recurrent layer, which takes an input of size 40 and has a hidden size of 40, followed by a single linear layer.

---

[8]Here, significant is defined to be p-value less than 0.05, but the results are not sensitive to this precise threshold.

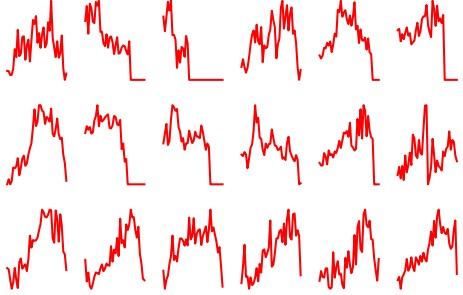

Figure C1: Fitted clathrin amplitudes for a few example events.

To train the AWD wavelet, the same warm start strategy was employed as in Appendix B. The AWD filters were trained for 100 epochs with Adam optimizer with a learning rate of 0.001. The experiment was run multiple times with respect to the randomness of mini-batches in the training procedure. All experiments were run on an AWS instance of p3.16xlarge for a few days.

## C.1 Distilled scaling functions and wavelets

Here we show the best wavelets selected by cross-validation and the corresponding scaling functions for 5 different runs of the experiments. The results are stable across multiple runs, all capturing information about how rapid changes in the clathrin trace is useful for predicting the auxilin response.

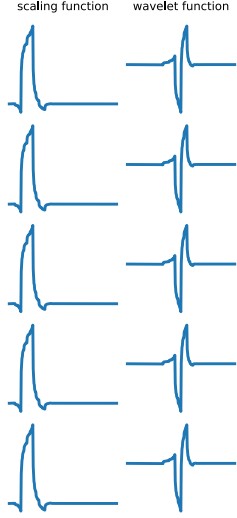

Figure C2: Optimal scaling and wavelet functions extracted by AWD across five random seeds.

## C.2 Varying sparsity and interpretation penalty

Fig C3 shows the learned wavelets distilled by AWD as the interpretation penalty $\gamma$ and the sparsity penalty $\lambda$ vary. Unlike Fig 4 where the lowpass filter is initialized to the DB 5 wavelet in the initial stage of training, here the lowpass filter is initialized to that corresponding to the Sym 5 wavelet. For large values of $\gamma$, the learned wavelets captures qualitatively the same biological features as those shown in Fig 4.

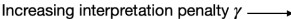

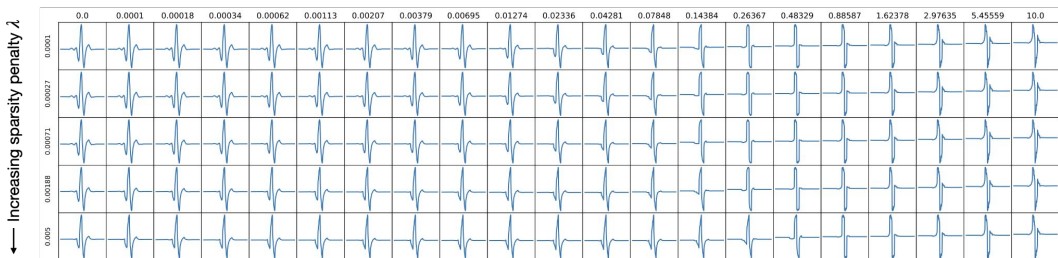

Figure C3: Varying sparsity and interpretation penalty yields different valid wavelets. In the initial stage of training, the lowpass filter is initialized to that corresponding to the Symlet 5 wavelet.

## C.3 Interpreting a single prediction

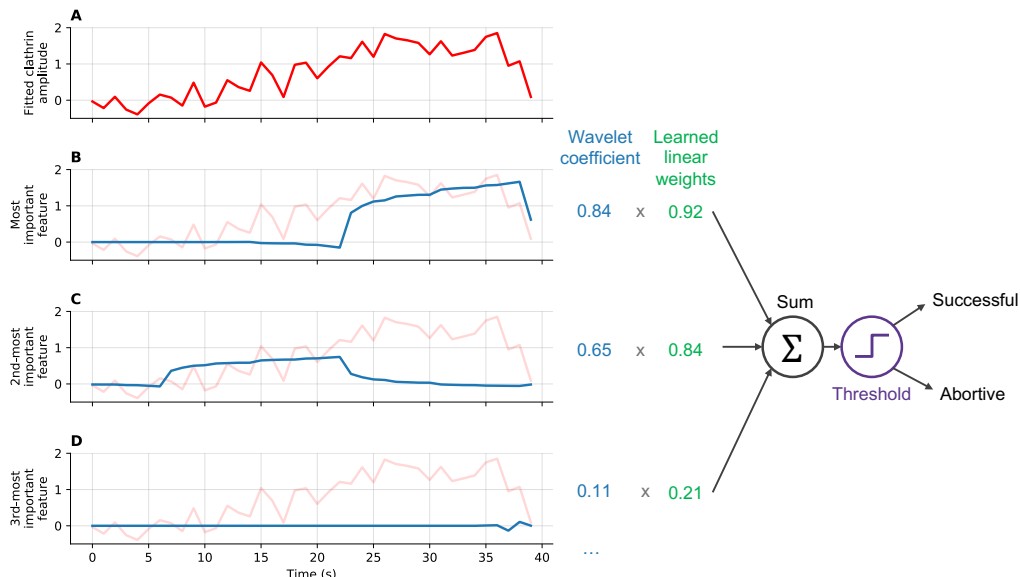

Figure C4: Interpreting a single prediction made by the wavelet model. The model takes the fitted clathrin amplitude shown in (**A**) and predicts that the event is successful. (**B, C, D**) show the three most important features for making this prediction. Each blue curve represents the input reconstruction for a single wavelet at a single scale. The curves in (**B**) and (**C**) seem to capture meaningful components of the clathrin signal, as they find a gradual rise in the signal, a large peak in the signal, and finally a steep drop in the signal at the end. The model is simply a linear combination of wavelet coefficients: each blue curve yields a coefficient which is then multiplied by a learned weight. The final prediction of successful or abortive is then made by thresholding the sum of these products. In this case, the first 2 coefficients dominate the prediction, and contributions for all remaining coefficients (some of which are omitted) are considerably less. For abortive predictions, the wavelet coefficients are usually much smaller (or negative).

## D Cosmological simulation details

For this task, we use the publicly available `MassiveNuS` simulation suite [62], composed of $101$ different $N$-body simulations spanning a range of cosmologies varying three parameters: the total neutrino mass $\Sigma m_\nu$, the normalization of the primordial power spectrum $A_s$, and the total matter density $\Omega_m$. These simulations are run at a single resolution of $1024^3$ particles for a $512$ Mpc/$h$ box size, and then ray-traced to obtain lensing convergence maps at source redshifts ranging from $z_s = 1.0$ to $z_s = 1100$. To build our dataset, we select 10 different cosmologies, listed in Table D1,

each of which provides $10,000$ mass maps at source redshift $z_s = 1$. We rebin these maps to size $256 \times 256$ with a pixel resolution of $0.8$ arcmin.

Table D1: Parameter values used in cosmology simulations.

| $m_\nu$ | $\Omega_m$ | $10^9 A_s$ |
|---|---|---|
| 0.0 | 0.3 | 2.1 |
| 0.06271 | 0.3815 | 2.2004 |
| 0.06522 | 0.2821 | 1.8826 |
| 0.06773 | 0.4159 | 1.6231 |
| 0.07024 | 0.2023 | 2.3075 |
| 0.07275 | 0.3283 | 2.2883 |
| 0.07526 | 0.3355 | 1.5659 |
| 0.07778 | 0.2597 | 2.4333 |
| 0.0803 | 0.2783 | 2.3824 |
| 0.08282 | 0.2758 | 1.8292 |

For training the AWD wavelet, we use the same warm start strategy as in Appendix B while the initial lowpass filter is initialized to the lowpass filter corresponding to the DB 5 wavelet. The AWD filters were trained for $50$ epochs with Adam optimizer with a learning rate of $0.001$. All experiments were run on an AWS instance of p3.16xlarge for a few days.

## D.1 Peak counting algorithm

Here we describe the peak counting algorithm developed in [59] to compare the performance of various filters. In weak lensing, peaks are defined as local maxima on the lensing convergence maps. In the original peak counting algorithm, a histogram is made for each convergence map based on counting the raw pixel (height) values of the peaks on the maps (see Fig D1). At training time, the mean histograms and the covariance matrices are then created for each setting of the cosmological parameters $\xi = (m_\nu, \Omega_m, 10^9 A_s)$; and at test time, individual histograms are compared to the mean histograms via the distance

$$d_{h,\xi} = (h - \mu_\xi)^\top \Sigma_\xi^{-1} (h - \mu_\xi),$$

and the parameters $\xi$ with the lowest distance $d_{h,\xi}$ is selected as prediction values. Here $h$ represents the histogram for a given map, and $\mu_\xi, \Sigma_\xi$, respectively, represent the mean histogram and the covariance matrix of the histograms for a cosmology with parameters $\xi$.

In [59], the peak counting algorithm is generalized to exploit more information around the peaks compared with the height of the peaks. Inspired by the first layer of the trained CNN for parameter estimation, they propose to use peak steepness based on the isotropic Laplace filter,

$$L = -\frac{10}{3} \begin{pmatrix} -0.05 & -0.2 & -0.05 \\ -0.2 & 1 & -0.2 \\ -0.05 & -0.2 & -0.05 \end{pmatrix},$$

which computes the difference of the peaks and the surrounding pixel values, or the Roberts cross kernels,

$$R_x = \begin{pmatrix} 0 & 1 \\ -1 & 0 \end{pmatrix}, R_y = \begin{pmatrix} 1 & 0 \\ 0 & -1 \end{pmatrix},$$

which compute the gradient at the peaks. For the Laplace filter, the peak steepness values are calculated via convolving the filter with the input images at the position of the peaks. For the Roberts cross kernels, the two filters $R_x$ and $R_y$ are applied to the $4$ adjacent $2 \times 2$ pixel blocks around the peaks and the magnitudes are calculated via $G_i = \sqrt{G_{x,i}^2 + G_{y,i}^2}, i = 1, \ldots, 4$, where $G_{x,i}$ and $G_{y,i}$ are the sub-images after convolve $R_x$ and $R_y$ with the $i$-th adjacent pixel blocks. Then the sum of the $4$ magnitudes $\sum_{i=1}^{4} G_i$ is used to get the peak steepness values.

Here we further use the wavelet filters distilled by AWD as peak-finding filters in the peak counting algorithm. To match the size of the distilled AWD filters with that of the Laplace filter or Roberts cross

kernels, we extract $3 \times 3$ subfilters from the wavelet filters where a majority of the mass is concentrated on. This results in $4$ different $3 \times 3$ filters, corresponding to three wavelet filters (LH,HL,HH) and one approximation filter (LL), which are then used as peak-finding filters to calculate the histograms of the peak steepness values. Fig D1 shows the distributions of peak steepness values using various filters mentioned above.

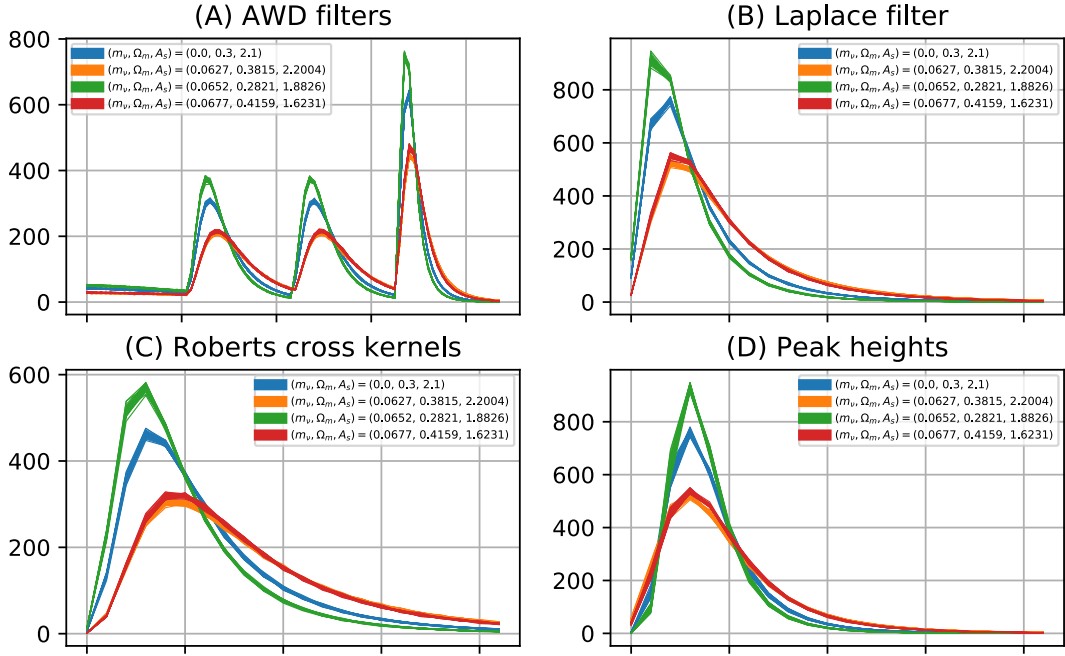

Figure D1: Peak steepness distributions using various filters.

To run the peak-counting algorithm with various filters, we need to select the number, width, and range of bins. For the Laplace filter and Roberts cross kernels, we use the same settings as [59] which runs bins from $0$ to $0.22$ in $0.01$ wide. In the case of the wavelet filters, we keep the same number of bins while the range is chosen via the algorithm's performance on a held-out validation set. The resulting bin is then used to evaluate the prediction performance on the test set.

## D.2 Wavelet activation maps

As part of our interpretability analysis, we now show images that highlight important features for predicting $\Omega_m$ (total fraction of matter in the universe) in Fig D2. To create the images, for each map we calculate feature attributions on the wavelet domain extracted by AWD using TRIM (here we use IG [37] to get attributions). Then only the wavelet coefficients with top $600$ attributions (out of $73,839$) are retained to transform back to the image domain using inverse wavelet transform. We can see that the activation maps highlight localized regions in the original maps that correspond to the high intensity peaks and voids. This is consistent with the known cosmology theory that these peaks contain high constraining power to predict cosmological parameters of the universe.

In the rightmost column of Fig D2, we also generate similar activation maps using the feature attributions in the pixel domain, where only the pixels with top $600$ attributions (out of $65,536$) are used. Similar to the wavelet activation maps, the activation maps in the pixel domain highlight isolated locations corresponding to the high intensity peaks. However, they fail to capture void (dark) regions at different scales and the maps only highlight the discrete number of pixels in the peaks and void regions—intuitively pixels in adjacent locations are competing with each other and the activation maps in the feature space will only select one of them as important. Indeed, we found that more than $10,000$ pixels are needed in the pixel domain to produce activation maps of similar quality as in the wavelet domain. Hence, AWD can provide qualitatively similar activation maps using far fewer features than the input domain.

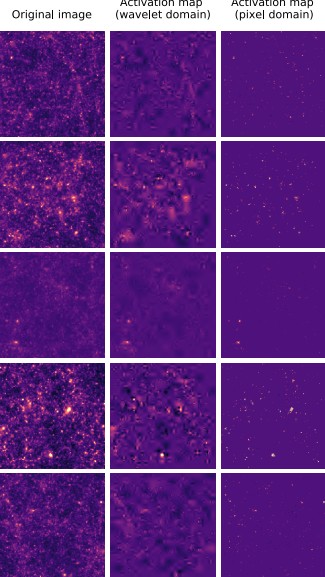

Figure D2: Sample images and activation maps using Integrated Gradients (IG). Left: 5 different weak gravitational lensing convergence maps selected randomly. Middle: wavelet activation maps for individual images made by the AWD model. Right: activation maps for individual images in the pixel domain.

Fig D3 below displays activation maps using the saliency TRIM attributions. Unlike the activation maps using IG, here we use wavelet coefficients / pixels with top $10,000$ attributions to generate activation maps. While the activation maps using saliency similarly highlight localized regions corresponding to the high intensity peaks and voids, the results are noisier and more number of features are required compared to the activation maps using IG.

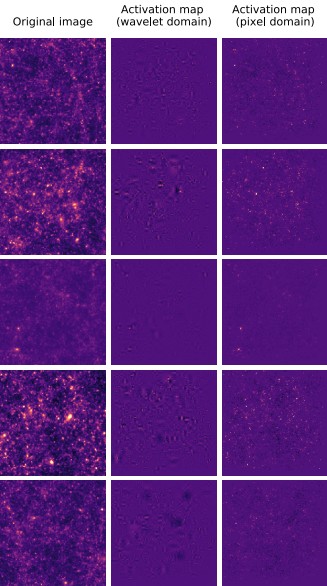

Figure D3: Sample images and activation maps using Saliency interpretations. Left: 5 different weak gravitational lensing convergence maps selected randomly. Middle: wavelet activation maps for individual images made by the AWD model. Right: activation maps for individual images in the pixel domain.

# E   MNIST

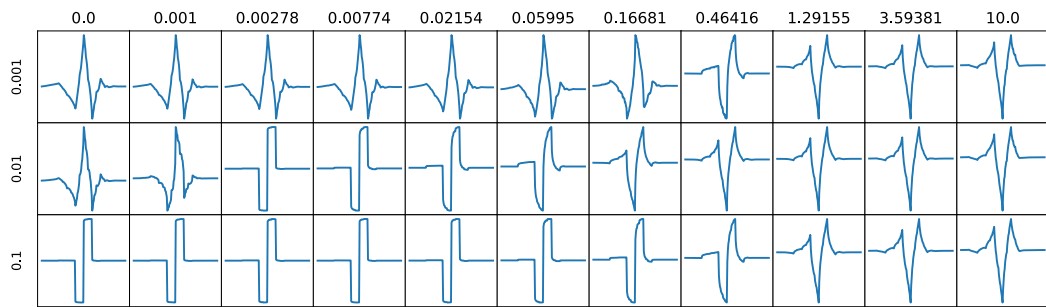

Figure E1: Varying sparsity (left) and interpretation (top) penalty yields dramatically different valid wavelets on the MNIST dataset. Increasing the sparsity penalty leads the wavelet to approach the well-known Haar wavelet function. Note: interpretation penalty is calculated only over the class "6".