# OpenReview forum: "Adaptive wavelet distillation from neural networks through interpretations"
_NeurIPS.cc/2021/Conference — NeurIPS 2021 Poster_

### Official Review · Reviewer_KLEM · 2021-07-15

**Rating:** 6
**Confidence:** 2

**Summary:**

The authors proposed Adaptive Wavelet Distillation (AWD) for a learnable wavelet transform. It distills the discriminative power of a deep neural network (DNN) by penalizing its feature attributions in the wavelet domain. Specifically, the authors proposed a new loss term consisting of reconstruction loss for invertibility, wavelet loss for yielding a valid transform, and interpretation loss for distilling the information of a trained DNN. Through synthetic, cosmology, and biological datasets, they demonstrated that the proposed method provides SoTA predictive performance and identifies interpretable features that are scientifically meaningful in the corresponding domains.

**Ethical Concerns:**

I do not see any ethical issues with this paper.

**Limitations And Societal Impact:**

The main limitation of the paper is that the claimed interpretability is not clearly demonstrated through the experiments. Please refer to the Main Review for more details.

**Main Review:**

(1) Originality
- The authors proposed to learn wavelet transforms based on the attributions from a trained DNN. To the best of my knowledge, this is a novel approach to use the wavelet transforms and knowledge distillation to interpret a trained DNN model.

(2) Quality
- The authors argue that wavelet models may or may not have better interpretability depending on a problem's context (line 36-38, line 121-123). Can you provide examples where wavelet models have better interpretability? Furthermore, I think explaining why having an orthogonal basis, and sparse representations indicate better interpretability would help understand the benefit of this paper.
- I am not sure the compression rate (line 243-245, 297-299) can show the better interpretability of the proposed method. In addition, while the authors stated that they had close collaboration with domain experts (lines 10-12), I could not find relevant details in the paper. Simply stating that the learned wavelet resembles a canonical pattern does not provide solid grounds.
- The interpretation loss induces the wavelet transform to yield the wavelet whose TRIM attribution is sparse concerning a trained DNN model. Since it largely deviates from the standard knowledge distillation setting where prediction outputs of a DNN are utilized, please provide a more detailed rationale for how the interpretation loss enables "gain information about the predicted outputs."
- It is unclear how critical the interpretation loss is for the proposed learnable wavelet transform. The authors only showed how the learned wavelet function changes in terms of the interpretation penalty (Figure 4). Please provide ablation studies showing how interpretation loss affects the predictive power of AWD. I think AWD without the interpretation loss may provide similar predictive performances and compression rates, which would indicate the distillation of a trained DNN is unnecessary.
- How critical are the initialization using known wavelet filters for the predictive performance? Is it more essential than the interpretation loss? Please compare with the results of using random initialization. Furthermore, please provide more detailed explanations for the different initialization methods (support sizes, Coiflet2, DB5).
- The authors claim that AWD outperforms the SoTA neural networks. Are the compared LSTM, and ResNet models are SoTA models? If not, please provide and adequately compare with the SoTA methods.
- The authors claim that AWD allows us to look at wavelet activation maps (Figure D2, line 307). However, activation maps can also be generated using the feature attribution in the input domain. Please show that whether AWD activation maps provide "better" interpretation. In addition, while the manuscript used a saliency map for attribution scores, Figure D2 uses integrated gradients. The inconsistency may lead to misunderstanding.

(3) Clarity
In general, the manuscript is clearly written. Another round of proofreading would be helpful, considering that there are some typos. (Ex. Line 79 (L_2), Line 308 Fig. C2).

(4) Significance
Although the paper has its own merits, unfortunately, it seems that both additional quantitative and qualitative analyses are required to claim better interpretability. While I agree that interpretability can be better defined in a particular problem, the manuscript does not seem to define and show the interpretability in the contexts properly. I think it does not fold great significance in the current form.


**Time Spent Reviewing:**

5

---

> ### Author Response · Authors · 2021-08-09
> **Thanks for your comments**
>
> Thanks for the helpful comments and thoughtful feedback (and time commitment!). We greatly appreciate your critique and address some of your concerns below.
>
> > Can you provide examples where wavelet models have better interpretability?
>
> Thanks for the question - indeed, there is a large crop of such examples that we only briefly hint at in the paper: we will revise the paper to better communicate them clearly. Both main examples in the paper should showcase where wavelet models have better interpretability. Specifically, they have a multiscale structure which allows interpreting the contribution of inputs to the final prediction at different locations and scales through a single mother wavelet; many domains in signal processing which can benefit from wavelet analysis share this property. This allow the final models to be extremely concise while still achieving state-of-the-art accuracy (details we will add to the paper): the biology model contains only 10 parameters (visualized in the wavelet function), along with only 30 coefficients in the sparse linear model and the final distilled cosmology model learns only 10 parameters. This allows a simple visualization of the wavelet function to understand the whole model.
>
> > Furthermore, I think explaining why having an orthogonal basis, and sparse representations indicate better interpretability would help understand the benefit of this paper.”
> >
> > “I am not sure the compression rate (line 243-245, 297-299) can show the better interpretability of the proposed method.”
>
> Thanks for pointing these out, we agree that we did not address them clearly in the paper. We will clarify these points in the introduction and section 4.2.
>
> - Having an orthogonal basis allows us to decompose the input signals into parts which contain information about the input independently (particularly different locations and different scales of the signal). The lack of information overlap between coefficients enables clearer interpretation; without this property, it would be very difficult to interpret the wavelet function itself, as one must simultaneously consider the context of many different overlapping filters. For example, in the pixel domain we much prefer to interpret an image in the standard basis representation (e.g. [1,0] and [0,1]) instead of interpreting in a dependent basis, such as [1, 1] and [-1, 0.5].
> - Having sparse representations / a higher compression rate is useful for many reasons. In terms of interpretability, it makes it easier to audit the non-zero coefficients which contribute to individual predictions. In addition, it makes downstream analysis, such as dimensionality reduction for visualization more tractable. Of course, compression is of independent interest for both computational speed and limiting memory consumption, which we can revise the text to point out more directly.
>
> > In addition, while the authors stated that they had close collaboration with domain experts (lines 10-12), I could not find relevant details in the paper.
>
> Good point! We will add the relevant details. The domain experts were intimately involved in generating / curating / processing the data as well as model development. One place this is particularly evident (besides the interpretations of the learned wavelets) is in the final model form used for different examples (i.e. the right-hand-side of Fig 2). For the biology problem a linear model is built on the top coefficients from different scales whereas for the cosmology problem the wavelet is used to count peaks in the input - both of these model forms came from knowledge about the domain problem.
>
> > please provide a more detailed rationale for how the interpretation loss enables "gain information about the predicted outputs."
>
> Indeed, the interpretation loss ensures that AWD produces model coefficients which are sparse in terms of their feature importances for predicting an outcome (i.e. the TRIM attribution). The intuition behind this penalty is that it allows for building an extremely simple model on top of these coefficients (e.g. a sparse linear model) which predicts well. This is only possible because the feature importances contain information about the predicted outputs - they convey how the model’s predicted output depends on the wavelet coefficients.
>
> > It is unclear how critical the interpretation loss is for the proposed learnable wavelet transform. The authors only showed how the learned wavelet function changes in terms of the interpretation penalty (Figure 4).
>
> Very good point! Here are numbers which show that the interpretation loss is crucial for improving performance:
>
> Table 1 molecular-partner prediction $R^2$ values (larger is better)
>
> | AWD | **AWD (no interpretation loss)** | Baseline wavelet | LSTM |
> | - | - | - | - |
> | 0.262 | **0.231**                                   | 0.197                  | 0.237 |
>
>
> Table 2 cosmology RMSE values (smaller is better)
>
> | AWD | **AWD (no interpretation loss)** | Baseline wavelet | Resnet |
> | - | - | - | - |
> |1.029 |  **1.354**                                    | 1.569                    | 1.156 |
>
>
> In both cases, AWD with the interpretation loss performs best. Adaptive wavelets without the interpretation loss still outperform the baseline wavelet, but fail to outperform the neural network models.
>
> > How critical are the initialization using known wavelet filters for the predictive performance? Is it more essential than the interpretation loss? Please compare with the results of using random initialization. Furthermore, please provide more detailed explanations for the different initialization methods (support sizes, Coiflet2, DB5).
>
> The interpretation loss certainly is more essential than the initialization. This is easiest seen in the simulation example (Fig 3). In this case, the only information that guides the adapted wavelets towards the groundtruth wavelets comes from the interpretation loss, since the wavelets are initialized to different well-known wavelets, which already satisfy the wavelet constraints. Here, the different initializations are DB5 + uniform noise (support size 10) , Sym5 (support size 10), and Coiflet2 (support size 12). Despite having different support sizes, the different initializations still all converge to be very similar to the groundtruth wavelet, thanks to the interpretation loss.
>
> Initializing to a known wavelet filter makes the optimization faster and more stable. The predictive performance can be unstable when initializing randomly (we ran some experiments on the mnist/cosmology/biology data), especially when the variance of the initialization is high. However, this can be solved by initializing at any known wavelet (and adding random noise to it) or first performing a few steps of the optimization without the interpretation loss before adding it in.
>
> > The authors claim that AWD outperforms the SoTA neural networks. Are the compared LSTM, and ResNet models are SoTA models?
>
> Yes, we apologize for the confusion - they are the SoTA models - we will make this clear in the text.
>
> > activation maps can also be generated using the feature attribution in the input domain. Please show whether AWD activation maps provide "better" interpretation. In addition, while the manuscript used a saliency map for attribution scores, Figure D2 uses integrated gradients. The inconsistency may lead to misunderstanding.
>
> Thanks for raising this point. The main benefit of interpretation by AWD goes beyond generating activation maps, but activation maps can be useful too, and we agree that the phrasing in Fig D2 failed to clearly convey the main advantage of getting the activation maps using AWD. As you pointed out, we can generate similar activation maps using the feature attribution in the pixel domain. There are two key advantages of using AWD here:
>
> 1. AWD induces sparse activation maps: since AWD produces the sparse TRIM attribution, only a few coefficients are important concerning the model’s outputs. In this example, only 600 most important wavelet coefficients were used out of 73,839 to generate these activation maps while we found more than 10,000 pixels were needed in the pixel domain to produce activation maps of similar quality. Hence we can gain qualitatively similar activation maps using far fewer features.
> 2. AWD produces more accurate activation maps. Even for activation maps with 10,000 pixels, they were not able to accurately capture some important features on the image. In particular, while the activation maps contained most of the localized high intensity peaks, they failed to capture void (dark) regions at different scales and the maps only highlighted the discrete number of pixels in the void regions - intuitively pixels in adjacent locations are competing with each other and the activation maps in the feature space will only select one of them as important. We will add this comparison to the paper and clarify it for the revision.
>
> Regarding your second concern, we apologize for the confusion. We observed that sparse attributions on the saliency map generally lead to the sparse attributions of the integrated gradients so we displayed them using the integrated gradients. We will add the activation maps using saliency and clarify it in the revised paper.
>
> > In general, the manuscript is clearly written. Another round of proofreading would be helpful, considering that there are some typos. (Ex. Line 79 (L_2), Line 308 Fig. C2).
>
> Thanks, we have fixed these and will do another proofread!
>
> > The main limitation of the paper is that the claimed interpretability is not clearly demonstrated through the experiments.
>
> We hope our answers above (especially the new numbers regarding the interpretation loss ablation study and model conciseness) helped show how the proposed approach can improve interpretability in real-world problems. Please let us know if you still have concerns in our response/ if there’s anything we can do to address them!

---

> > ### Comment · Reviewer_KLEM · 2021-08-29
> > **Post-Rebuttal**
> >
> > Thank you for the detailed feedback on my comments. Although I still have concerns about the claimed interpretability, I also think that the quality of the manuscript would be increased if revised according to the authors' feedbacks. Therefore, I raised my rating to 6.

---

### Official Review · Reviewer_XipH · 2021-07-16

**Rating:** 7
**Confidence:** 3

**Summary:**

The authors propose an extended approach for learnable wavelet transforms, and add a distillation-based regularization term to the loss, which penalizes parts of the wavelet with low importance scores in the wavelet-transformed space given a pre-trained neural network.
In experiments on synthetic data, as well as two scientific datasets, their predictive performance is quantitatively analyzed.
Additionally, the interpretability of the learned wavelets is analyzed qualitatively.


**Limitations And Societal Impact:**

Limitations are not discussed.
While wavelets provide a powerful framework, the authors should discuss in which cases the training may fail.
The problem of the generally loose term of "explanation" as discussed in the introduction.
No negative societal impact is reported.
Maybe at least a potential societal impact more general terms of interpretability may be addressed.
For example, over-confidence in a model caused by over-interpretation of possible red herrings in the wavelets could have a negative impact.

**Main Review:**

### Originality:
Previous approaches to learned sparse wavelet transforms are extended by additional constraints from previous literature.
The main originality is the proposed neural network-attribution-based distillation regularization term introduced into the learned wavelet loss.
The paper is fairly original, and differentiates itself well from previous work.

### Quality:
It is not clear, how much the teacher model actually contributes to the increased performance with the intepretation loss, especially in Section 4.3, where the AWD outperforms its teacher.
Reporting the RSME for the case where the interpretation penalty is set to zero in Tables 1 and 2 may give insight to whether and how much the interpretation contributes to increased performance.

The authors claim high interpretability of the learned wavelets.
This may be highly dependent on the problem domain.
Introducing an experiment for which wavelets may not be the best choice, may better show the limitations of the interpretability of method, and make it more convincing.
While it is intuitive, that these learned wavelets give a much clearer idea of their produced predictions, the analysis to support the claim is purely qualitative, and may benefit from a more quantitative analysis.
Such an analysis would furthermore improve the quality for the paper, since the two real-world experiments rely on expert knowledge for their qualitative interpretation.


### Clarity:
The paper is clearly written, the introduced methods are introduced well and sufficiently.

Figure 4 shows qualitatively how different hyperparameters yield different wavelets.
Beyond the fact that they change, it is not really clear to me how the different wavelets reflect the prediction strategy.
A discussion of the possibly increased intuitive "goodness" of the wavelets for different parameters may increase the clarity of this experiment

Minor:
- l.121-122 may or not -> may or may not

### Significance:
Although a broader application of the method may increase its significance, the results suggest a valid extension for learnable wavelets.




**Time Spent Reviewing:**

4

---

> ### Author Response · Authors · 2021-08-07
> **Thanks for your comments**
>
>
> Thanks for the helpful comments and thoughtful feedback (and large time commitment!). We sincerely appreciate your critique and address some of your concerns below.
>
> > It is not clear, how much the teacher model actually contributes to the increased performance with the intepretation loss, especially in Section 4.3, where the AWD outperforms its teacher. Reporting the RSME for the case where the interpretation penalty is set to zero in Tables 1 and 2 may give insight to whether and how much the interpretation contributes to increased performance.
>
> Very good point! Here are those numbers when removing the interpretation loss. In both cases, AWD with the interpretation loss performs best. Adaptive wavelets without the interpretation loss still outperform the baseline wavelet, but fail to outperform the neural network models.
>
> Table 1 molecular-partner prediction $R^2$ values (larger is better)
>
> | AWD | **AWD (no interpretation loss)** | Baseline wavelet | LSTM |
> | ------- | -------------------------------------- | ---------------------- | -------- |
> | 0.262 | **0.231**                                   | 0.197                  | 0.237 |
>
>
> Table 2 cosmology RMSE values (smaller is better)
>
> | AWD | **AWD (no interpretation loss)** | Baseline wavelet | Resnet |
> | ------- | -------------------------------------- | ---------------------- | -------- |
> |1.029 |  **1.354**                                    | 1.569                    | 1.156 |
>
>
> In the case of the simulated data (Fig 3) this is even more clear. The only information that guides the adapted wavelets towards the groundtruth wavelets comes from the interpretation loss, since the wavelets are initialized to different well-known wavelets, which already satisfy the wavelet constraints.
>
> > The authors claim high interpretability of the learned wavelets. This may be highly dependent on the problem domain. Introducing an experiment for which wavelets may not be the best choice, may better show the limitations of the interpretability of method, and make it more convincing.
>
> Certainly true - we have recently run an example or two which fit this description which we can add to the supplement. One example is MNIST, where changing the interpretation penalties yields qualitatively different shapes for the wavelet function, but any reasonable wavelet function yields comparable predictive performance (likely because the task can be solved with relatively coarse features which do not require finely tuned wavelets). Nevertheless, increasing the interpretation penalty interestingly leads to recovering wavelet functions closely resembling Haar wavelets, which are able to compress the input representation well.
>
> > While it is intuitive, that these learned wavelets give a much clearer idea of their produced predictions, the analysis to support the claim is purely qualitative, and may benefit from a more quantitative analysis. Such an analysis would furthermore improve the quality for the paper, since the two real-world experiments rely on expert knowledge for their qualitative interpretation.
>
> We agree that ideally we could have more metrics to quantify the interpretability. We can partially address this by adding details about the simplicity of the final models to the main text and about their interpretation to the supplement. For example, the final parameter counts for the models is currently missing: the biology model contains only 10 parameters (visualized in the wavelet function), along with only 30 coefficients in the sparse linear model. The final distilled cosmology model learns only 10 parameters to make its predictions. We hope that this extreme frugality of parameters, along with the properties of wavelets (e.g. orthogonal, multiscale features), and the vetting by domain experts serves as sufficient evaluation of the interpretability of the wavelet models.
>
> > Figure 4 shows qualitatively how different hyperparameters yield different wavelets. Beyond the fact that they change, it is not really clear to me how the different wavelets reflect the prediction strategy. A discussion of the possibly increased intuitive "goodness" of the wavelets for different parameters may increase the clarity of this experiment
>
> Thanks for pointing this out! Indeed it is difficult to tell from the figure without domain knowledge how different wavelets alter the predictions. We have created and will add a supplementary figure which illustrates precisely which parts of the input contribute to the coefficients used to make very positive and very negative predictions. For the wavelets in Fig 4, it becomes clear that the key to a “good” wavelet for this problem is that the wavelet function has precisely two big dips in opposite directions (corresponding to a biologically meaningful creation and dissipation of clathrin-coated pits), unlike the wavelets in the top-left which have several smooth dips, and in the bottom right which have three big dips.
>
> > l.121-122 may or not -> may or may not
>
> Thanks for catching this! We have fixed it for the revision.
>
> > Limitations are not discussed. While wavelets provide a powerful framework, the authors should discuss in which cases the training may fail. The problem of the generally loose term of "explanation" as discussed in the introduction. No negative societal impact is reported. Maybe at least a potential societal impact more general terms of interpretability may be addressed. For example, over-confidence in a model caused by over-interpretation of possible red herrings in the wavelets could have a negative impact.
>
> Thanks for these points! Indeed, we will clarify the cases where wavelet models perform well (i.e. structured signals, usually in the time/space domains) and be more precise about what we mean by the word “interpretation”. We will also add in some potential negative social impacts - thanks for giving us the over-confidence example.

---

> > ### Comment · Reviewer_XipH · 2021-08-27
> > **Satisfied, increased rating**
> >
> > Thank you for the detailed replies to all reviewers' comments.
> > I am satisfied with the new additions to the manuscript and clarifications.
> > As I feel the quality of the manuscript increased, I raise my rating to 7.

---

### Official Review · Reviewer_HEh3 · 2021-07-16

**Rating:** 7
**Confidence:** 4

**Summary:**

This paper proposes learning an informative wavelet decomposition of the input space of a neural network that is able to come up with sparse attributions of the features used in the output. The idea is that in some scientific domains which collect time or space oriented data (in the form of images or signals), the features that lead to a particular prediction are hard to interpret because of complex patterns, and that wavelets can instead provide a concise description since they are localized in both frequency and space.

**Limitations And Societal Impact:**

The authors should clearly set out the use case for this contribution. Clearly this is only useful for signals in time or space domains.

**Main Review:**

Overall, I find this to be a valuable contribution in overcoming the black boxness of neural networks used in scientific domains. The example case studies in biology and cosmology are convincing and clearly written even to those with a limited background in this field.  The summary of the wavelet transform (S2.1) is clear enough to make this paper accessible to a reader who is unfamiliar with wavelets.  However, the overall way the paper is written could be improved.  First, the authors should make it clear what sort of domains benefit from these---clearly structured domains. For example frequency domain features would likely be of little use in interpreting DNA sequences or text data. It also probably more useful in multi-granular settings where despite the high dimensionality of the underlying space, the phenomenon is a signal over the space. From this standpoint, this work could probably also be extended to graph signals.

Second, the authors don't use much bandwidth to discuss selection of wavelets, scales, or levels (in cases of multilevel transforms), but these are precisely what would be domain-specific, as the authors note that interpretations often are. Indeed it seems as if first learning a useful dictionary of wavelets may be helpful. See Aharon et al. 2006, or Ophir et al. 2011 for wavelet dictionary learning algorithms.

Figure 1 does not seem very helpful. The authors instead should showcase that the transform is trained with several losses including an autoencoder-like loss reconstruction loss, and a sparsity loss which the authors call 'interpretability loss.' This could be visualized along side a neural network.

In Line 43, the statement that wavelets have not been used for interpretability is not quite true. I believe there are quite a few papers applying wavelet and using it in terms of model interpretability. Just a few sample papers:
Wang et al., Multilevel Wavelet Decomposition Network for Interpretable Time Series Analysis
Li et al, WaveletKernelNet: An Interpretable Deep Neural Network for Industrial Intelligent Diagnosis
Khan et al, Learning filter widths of spectral decompositions with wavelets

Additional points of clarification:

1) Do the wavelets have to be initialized to a known filter or it can be random? Is the distance loss added during training to recover the ground truth wavelet?

2) In the experimental section, the authors showed the AWD method helped enhance the prediction accuracy, but how is the interpretability compared to other methods? Can the authors provide some comparisons of AWD and other methods so the idea of enhancing interpretability is further illustrated?

I would also suggest the authors to conduct an ablation study to see the effects of each loss term on learning the wavelet transforms? Since the key of this method is the combination of three losses, it’s good to show how these loss terms affect the model.

I am satisfied by the author responses and will maintain my high score.


**Time Spent Reviewing:**

4

---

> ### Author Response · Authors · 2021-08-07
> **Thanks for your comments**
>
> Thanks for the helpful comments and thoughtful feedback (and large time commitment!). We sincerely appreciate your critique and address some of your concerns below.
>
> > First, the authors should make it clear what sort of domains benefit from these---clearly structured domains. For example frequency domain features would likely be of little use in interpreting DNA sequences or text data. It also probably more useful in multi-granular settings where despite the high dimensionality of the underlying space, the phenomenon is a signal over the space. From this standpoint, this work could probably also be extended to graph signals.
>
> Very good point, certainly wavelet models do not work well in all cases and we will clarify which domains can benefit from our approach (i.e. many domains in signal processing which possess multis-scale structure). We also agree that interpreting DNA sequences or text data in the frequency domain would not be so useful and less interpretable. Extending this work to graph signals sounds very interesting!
>
> > Second, the authors don't use much bandwidth to discuss selection of wavelets, scales, or levels (in cases of multilevel transforms), but these are precisely what would be domain-specific, as the authors note that interpretations often are. Indeed it seems as if first learning a useful dictionary of wavelets may be helpful. See Aharon et al. 2006, or Ophir et al. 2011 for wavelet dictionary learning algorithms.
>
> Indeed this is a tricky problem - in this work we used domain knowledge to pick reasonable choices for these parameters, and after learning the wavelet function found the performance to be surprisingly good without tuning these parameters; we will devote some text to contextualizing these results. Thanks for the references, they are certainly related and we will include them - they could certainly provide good initializations for the wavelet functions learned here!
>
> > Figure 1 does not seem very helpful. The authors instead should showcase that the transform is trained with several losses including an autoencoder-like loss reconstruction loss, and a sparsity loss which the authors call 'interpretability loss.' This could be visualized along side a neural network.
>
> Thanks for the feedback! We will adapt the figure to show and emphasize these losses.
>
> > In Line 43, the statement that wavelets have not been used for interpretability is not quite true. I believe there are quite a few papers applying wavelet and using it in terms of model interpretability. Just a few sample papers: Wang et al., Multilevel Wavelet Decomposition Network for Interpretable Time Series Analysis Li et al, WaveletKernelNet: An Interpretable Deep Neural Network for Industrial Intelligent Diagnosis Khan et al, Learning filter widths of spectral decompositions with wavelets
>
> You are correct! We will restrict the statement to be more specific to using wavelets for interpretable model distillation. Also thanks for the refs (we will certainly add them).
>
> > Do the wavelets have to be initialized to a known filter or it can be random?
>
> Initializing to a known wavelet filter makes the optimization faster and more stable. The predictive performance can be unstable when initializing randomly (we just ran some experiments on the mnist/biology data), especially when the variance of the initialization is high. However, this can be solved by initializing at any known wavelet (potentially with some random noise added to it) or first performing a few steps of the optimization without the interpretation loss before adding it in.
>
> > Is the distance loss added during training to recover the ground truth wavelet?
>
> The distance loss is not added during training to recover the ground truth! This showcases how the information from the gradients of the trained neural network alone are sufficient to recover the ground truth wavelet.
>
> > In the experimental section, the authors showed the AWD method helped enhance the prediction accuracy, but how is the interpretability compared to other methods? Can the authors provide some comparisons of AWD and other methods so the idea of enhancing interpretability is further illustrated?
>
> In terms of distilling into interpretable models, existing distillation approaches lead to model forms (e.g. GAMs or decision trees) which have much poorer predictive performance on these data problems. This leaves post-hoc interpretability methods, where the dominant approach is to generate saliency maps for individual model predictions. We will add some saliency maps for popular techniques alongside Fig D2 to show how they compare. However, it is worth noting that these methods have suffered from many issues (e.g. see “Sanity Checks for Saliency Maps” (Adebayo et al. 2018)), with no clear consensus on how to deal with problems such as scoring interactions between time points/pixels in a signal. In contrast, interpretations of the wavelet models are much more straightforward, as the predictions are a simple function of its inputs and thus don’t possess the ambiguities present in summarizing the interactions of a neural network.
>
> > I would also suggest the authors to conduct an ablation study to see the effects of each loss term on learning the wavelet transforms? Since the key of this method is the combination of three losses, it’s good to show how these loss terms affect the model.
>
> Very good point! Here are those numbers when removing the interpretation loss. In both cases, AWD with the interpretation loss performs best. Adaptive wavelets without the interpretation loss still outperform the baseline wavelet, but fail to outperform the neural network models.
>
> Table 1 molecular-partner prediction $R^2$ values (larger is better)
>
> | AWD | **AWD (no interpretation loss)** | Baseline wavelet | LSTM |
> | ------- | -------------------------------------- | ---------------------- | -------- |
> | 0.262 | **0.231**                                   | 0.197                  | 0.237 |
>
>
> Table 2 cosmology RMSE values (smaller is better)
>
> | AWD | **AWD (no interpretation loss)** | Baseline wavelet | Resnet |
> | ------- | -------------------------------------- | ---------------------- | -------- |
> |1.029 |  **1.354**                                    | 1.569                    | 1.156 |

---

### Official Review · Reviewer_eZkr · 2021-07-20

**Rating:** 6
**Confidence:** 3

**Summary:**

The paper proposes distilling a DNN into a learnable wavelet transform. Producing an interpretable architecture, that was used in two scientific applications cosmology and molecular-partner prediction.
The paper proposed 3 losses: In AWD the wavelet transform should be invertible, allowing for reconstruction of the original data. This ensures that the transform does not lose any information in the input data "reconstruction loss".  The distillation part of AWD is added by calculating the attribution scores of a  given model for each coefficient in the wavelet representation and try to find a wavelet function that makes these attributions sparse "interpretation loss". Finally, the wavelet loss which ensures that the learned filters yield a valid wavelet transform.

The paper provides two scientific data problems where the resulting wavelet model is sufficiently interpretable for use and provides similar prediction performance as the DNN.  The paper also evaluated AWD on Synthetic data showing that AWD can recover groundtruth wavelets from noisy data.



**Main Review:**

Strength:
- The work is novel, and original.
- The paper is well written and clear.
- Significance of work is high since learned wavelets can be used as interpretations for scientific domains.
- The submission is technically sound.

Weakness:
- No comparison was done with other distillation methods.
- Transformation Importance relies on gradients for importance identifications which are noisy.

Questions:
- Do you have any insights on why distillation improves accuracy above DNN? Typically the upper bound is DNN accuracy.

Typos:
- line 106 bijective-> objective
- line 176 think you mean Fig 2?

**Time Spent Reviewing:**

4

---

> ### Author Response · Authors · 2021-08-06
> **Thank you for your comments**
>
> Thanks for the helpful comments and thoughtful feedback (and large time commitment!). We appreciate your critique and address some of your concerns below.
>
> > Transformation Importance relies on gradients for importance identifications which are noisy.
>
> This is a good point! It’s worth noting that transformation importance is quite general, and also supports importances that are not gradient-based, such as SHAP or Contextual Decomposition. We originally thought we would need to compute non gradient-based importances, but found that the simple gradient worked very well and didn’t want to needlessly add extra complexity to the method. It would certainly be interesting to explore other importance methods in future work.
>
> > No comparison was done with other distillation methods.
>
> Unfortunately, we couldn’t find comparable distillation methods that provided both strong predictive accuracy and interpretability. The other existing methods either do not provide interpretability (e.g. distill into another smaller neural network) or distill whose model form (e.g. decision tree or GAM) considerably degrades predictive performance. Instead, we compare with the state-of-the-art neural networks and other popular wavelet functions.
>
> > “Do you have any insights on why distillation improves accuracy above DNN? Typically the upper bound is DNN accuracy.”
>
> This was a surprise to us as well! Our intuition for this phenomenon is that for these problems, (adaptive) wavelets match the underlying structure of the problem well, allowing the simple, distilled model to predict well while limiting its potential for overfitting. This should apply to many problems that share this underlying structure, but certainly not to all problems.
>
> > line 106 bijective-> objective
> >
> > line 176 think you mean Fig 2?
>
> Thank you for catching those typos! (in line 106 we did actually mean bijective, as in the mapping is invertible)

---

> ### Comment · Reviewer_eZkr · 2021-08-26
> **Thank you**
>
> I would like to thank the authors for their response. My score remains unchanged.

---

### Author Response · Authors · 2021-08-26
**We are happy to answer more questions if there still exist concerns for our paper**

Dear Reviewers,

Thanks for your time and efforts in reviewing our paper. We appreciate your constructive comments. Hopefully, our response can address your concerns. If you have further questions or confusion, we would be very happy to clarify. Thank you very much.

Best,

Authors

---

### Decision · Program_Chairs · 2021-09-27

**Decision:**

Accept (Poster)

**Comment:**

The reviewers agreed that this paper had many strengths as a novel method. Presenting two applications as case studies was a welcome departure from standard NeurIPS submissions. The reviewers have given you a few actionable suggestions which I hope you will take for the final version. The idea of including examples where wavelets are not the best choice in the supplementals is a very good one.